# Fast Bayesian Inference with Batch Bayesian Quadrature via Kernel Recombination

**Masaki Adachi**[*]
Machine Learning Research Group, University of Oxford
Toyota Motor Corporation
masaki@robots.ox.ac.uk

**Satoshi Hayakawa**[*]**, Harald Oberhauser**
Mathematical Institute, University of Oxford
{hayakawa,oberhauser}@maths.ox.ac.uk

**Martin Jørgensen, Michael A. Osborne**
Machine Learning Research Group, University of Oxford
{martinj, mosb}@robots.ox.ac.uk

## Abstract

Calculation of Bayesian posteriors and model evidences typically requires numerical integration. Bayesian quadrature (BQ), a surrogate-model-based approach to numerical integration, is capable of superb sample efficiency, but its lack of parallelisation has hindered its practical applications. In this work, we propose a parallelised (batch) BQ method, employing techniques from kernel quadrature, that possesses an empirically exponential convergence rate. Additionally, just as with Nested Sampling, our method permits simultaneous inference of both posteriors and model evidence. Samples from our BQ surrogate model are re-selected to give a sparse set of samples, via a kernel recombination algorithm, requiring negligible additional time to increase the batch size. Empirically, we find that our approach significantly outperforms the sampling efficiency of both state-of-the-art BQ techniques and Nested Sampling in various real-world datasets, including lithium-ion battery analytics.[2]

## 1 Introduction

Many applications in science, engineering, and economics involve complex simulations to explain the structure and dynamics of the process. Such models are derived from knowledge of the mechanisms and principles underlying the data-generating process, and are critical for scientific hypothesis-building and testing. However, dozens of plausible simulators describing the same phenomena often exist, owing to differing assumptions or levels of approximation. Similar situations can be found in selection of simulator-based control models, selection of machine learning models on large-scale datasets, and in many data assimilation applications [28].

In such settings, with multiple competing models, choosing the best model for the dataset is crucial. Bayesian model evidence gives a clear criteria for such model selection. However, computing model evidence requires integration over the likelihood, which is challenging, particularly when the likelihood is non-closed-form and/or expensive. The ascertained model is often applied to

---

[*]Equal contribution
[2]Code: https://github.com/ma921/BASQ

36th Conference on Neural Information Processing Systems (NeurIPS 2022).

produce posteriors for prediction and parameter estimation afterwards. There are many algorithms specialised for the calculation of model evidences or posteriors, although only a limited number of Bayesian inference solvers estimate both model evidence *and* posteriors in one go. As such, costly computations are often repeated (at least) twice. Addressing this concern, nested sampling (NS) [71, 46] was developed to estimate both model evidence and posteriors simultaneously, and has been broadly applied, especially amongst astrophysicists for cosmological model selection [63]. However, NS is based on a Monte Carlo (MC) sampler, and its slow convergence rate is a practical hindrance.

To aid NS, and other approaches, parallel computing is widely applied to improve the speed of wall-clock computation. Modern computer clusters and graphical processing units enable scientists to query the likelihood in large batches. However, parallelisation can, at best, linearly accelerate NS, doing little to counter NS's inherently slow convergence rate as a MC sampler.

This paper investigates batch Bayesian quadrature (BQ) [65] for fast Bayesian inference. BQ solves the integral as an inference problem, modelling the likelihood function with a probabilistic model (typically a Gaussian process (GP)). Gunter et al. [37] proposed Warped sequential active Bayesian integration (WSABI), which adopts active learning to select samples upon uncertainty over the integrand. WSABI showed that BQ with expensive GP calculations could achieve faster convergence in wall time than cheap MC samplers. Wagstaff et al. [78] introduced batch WSABI, achieving even faster calculation via parallel computing and became the fastest BQ model to date. We improve upon these existing works for a large-scale batch case.

## 2  Background

**Vanilla Bayesian quadrature**    While BQ in general is the method for the integration, the functional approximation nature permits solving the following integral $Z$ and obtaining the surrogate function of posterior $p(x)$ simultaneously in the Bayesian inference context:

$$p(x) = \frac{\ell_{\text{true}}(x)\pi(x)}{Z} = \frac{\ell_{\text{true}}(x)\pi(x)}{\int \ell_{\text{true}}(x)\pi(x)\,\mathrm{d}x}, \tag{1}$$

where both $\ell_{\text{true}}(x)$ (e.g. a likelihood) and $\pi(x)$ (e.g. a prior) are non-negative, and $x \in \mathbb{R}^d$ is a sample, and is sampled from prior $x \sim \pi(x)$. BQ solves the above integral as an inference problem, modelling a likelihood function $\ell(x)$ by a GP in order to construct a surrogate model of the expensive true likelihood $\ell_{\text{true}}(x)$. The surrogate likelihood function $\ell(x)$ is modelled:

$$\ell \,|\, \mathbf{y} \sim \mathcal{GP}(\ell; m_{\mathbf{y}}, C_{\mathbf{y}}), \tag{2a}$$

$$m_{\mathbf{y}}(x) = K(x, \mathbf{X})K(\mathbf{X}, \mathbf{X})^{-1}\mathbf{y}, \tag{2b}$$

$$C_{\mathbf{y}}(x, x') = K(x, x') - K(x, \mathbf{X})K(\mathbf{X}, \mathbf{X})^{-1}K(\mathbf{X}, x'), \tag{2c}$$

where $\mathbf{X} \in \mathbb{R}^{n \times d}$ is the matrix of observed samples, $\mathbf{y} \in \mathbb{R}^n$ is the observed true likelihood values, $K$ is the kernel. [3] Due to linearity, the mean and variance of the integrals are simply

$$\mathbb{E}[Z \,|\, \mathbf{y}] = \int m_{\mathbf{y}}(x)\pi(x)\,\mathrm{d}x, \tag{3a}$$

$$\mathbb{V}\mathrm{ar}[Z \,|\, \mathbf{y}] = \iint C_{\mathbf{y}}(x, x')\pi(x)\pi(x')\,\mathrm{d}x\,\mathrm{d}x'. \tag{3b}$$

In particular, (3) becomes analytic when $\pi(x)$ is Gaussian and $K$ is squared exponential kernel, $K(\mathbf{X}, x) = v\sqrt{|2\pi\mathbf{W}|}\mathcal{N}(\mathbf{X}; x, \mathbf{W})$, where $v$ is kernel variance and $\mathbf{W}$ is the diagonal covariance matrix whose diagonal elements are the lengthscales of each dimension. Since both the mean and variance of the integrals can be calculated analytically, posterior and model evidence can be obtained simultaneously. Note that non-Gaussian prior and kernel are also possible to be chosen for modelling via kernel recombination (see Supplementary). Still, we use this combination throughout this paper for simplicity.

---

[3] In GP modelling, the GP likelihood function is modelled as $\mathcal{GP}(0, K)$, and (2) is the resulting posterior GP. Throughout the paper, we refer to a symmetric positive semi-definite kernel just as a kernel. The notations $\sim$ and $|$ refer to being sampled from and being conditioned, respectively.

**Warped Bayesian quadrature (WSABI)** WSABI with linearisation approximation (WSABI-L) adopts the square-root warping GP for non-negativity with linearisation approximation of the transform $\tilde{\ell} \mapsto \ell = \alpha + \frac{1}{2}\tilde{\ell}^2$. [4] The square-root GP is defined as $\tilde{\ell} \sim \mathcal{GP}(\tilde{\ell}; \tilde{m}_\mathbf{y}, \tilde{C}_\mathbf{y})$, and we have the following linear approximation:

$$\ell \mid \mathbf{y} \sim \mathcal{GP}(\ell; m_\mathbf{y}^L, C_\mathbf{y}^L), \tag{4a}$$

$$m_\mathbf{y}^L(x) = \alpha + \frac{1}{2}\tilde{m}_\mathbf{y}(x)^2, \tag{4b}$$

$$C_\mathbf{y}^L(x, x') = \tilde{m}_\mathbf{y}(x)\tilde{C}_\mathbf{y}(x, x')\tilde{m}_\mathbf{y}(x'). \tag{4c}$$

Gaussianity implies the model evidence $Z$ and posterior $p(x)$ remain analytical (see Supplementary).

**Kernel quadrature in general** kernel quadrature (KQ) is the group of numerical integration rules for calculating the integral of function classes that form the reproducing kernel Hilbert space (RKHS). With a KQ rule $Q_{\mathbf{w}, \mathbf{X}}$ given by weights $\mathbf{w} = (w_i)_{i=1}^n$ and points $\mathbf{X} = (x_i)_{i=1}^n$, we approximate the integral by the weighted sum

$$Q_{\mathbf{w}, \mathbf{X}}(h) := \sum_{i=1}^n w_i h(x_i) \approx \int h(x)\pi(x)\,\mathrm{d}x, \tag{5}$$

where $h$ is a function of RKHS $\mathcal{H}$ associated with the kernel $K$. We define its worst-case error by $\mathrm{wce}(Q_{\mathbf{w}, \mathbf{X}}) := \sup_{\|h\|_{\mathcal{H}} \leq 1} |Q_{\mathbf{w}, \mathbf{X}}(h) - \int h(x)\pi(x)\,\mathrm{d}x|$. Surprisingly, it is shown in [48] that we have

$$\mathbb{V}\mathrm{ar}[Z \mid \mathbf{y}] = \inf_{\mathbf{w}} \mathrm{wce}(Q_{\mathbf{w}, \mathbf{x}})^2. \tag{6}$$

Thus, the point configuration in KQ with a small worst-case error gives a good way to select points to reduce the integral variance in Bayesian quadrature.

**Random Convex Hull Quadrature (RCHQ)** Recall from (5) and (6) that we wish to approximate the integral of a function $h$ in the current RKHS. First, we prepare $n-1$ test functions $\varphi_1, \ldots, \varphi_{n-1}$ based on $M$ sample points using the Nyström approximation of the kernel: $\varphi_i(x) := u_i^\top K(\mathbf{X}_\mathrm{nys}, x)$, where $u_i \in \mathbb{R}^M$ is the $i$-th eigenvector of $K(\mathbf{X}_\mathrm{nys}, \mathbf{X}_\mathrm{nys})$. If we let $\lambda_i$ be the $i$-th eigenvalue of the same matrix, the following gives a practical approximation [55]:

$$K_0(x, y) := \sum_{i=1}^{n-1} \lambda_i^{-1} \varphi_i(x)\varphi_i(y). \tag{7}$$

Next, we consider extracting a weighted set of $n$ points $(\mathbf{w}_\mathrm{quad}, \mathbf{X}_\mathrm{quad})$ from a set of $N$ points $\mathbf{X}_\mathrm{rec}$ with positive weights $\mathbf{w}_\mathrm{rec}$. We do it by the so-called kernel recombination algorithm [59, 75], so that the measure induced by $(\mathbf{w}_\mathrm{quad}, \mathbf{X}_\mathrm{quad})$ exactly integrates the above test functions $\varphi_1, \ldots, \varphi_{n-1}$ with respect to the measure given by $(\mathbf{w}_\mathrm{rec}, \mathbf{X}_\mathrm{rec})$ [44].

In the actual implementation of multidimensional case, we execute the kernel recombination not by the algorithm [75] with the best known computational complexity $\mathcal{O}(C_\varphi N + n^3 \log(N/n))$ (where $C_\varphi$ is the cost of evaluating $(\varphi_i)_{i=1}^{n-1}$ at a point), but the one of [59] using an LP solver (Gurobi [38] for this time) with empirically faster computational time. We also adopt the randomized SVD [40] for the Nyström approximation, so we have a computational time empirically faster than $\mathcal{O}(NM + M^2 \log n + Mn^2 \log(N/n))$ [44] in practice.

## 3 Related works

**Bayesian inference for intractable likelihood** Inference with intractable likelihoods is a long-standing problem, and a plethora of methods have been proposed. Most infer posterior and evidence separately, and hence are not our fair competitors, as solving both is more challenging. For posterior inference, *Markov Chain Monte Carlo* [62, 43], particularly *Hamilton Monte Carlo* [47], is the gold standard. In a *Likelihood-free inference* context, kernel density estimation (KDE) with Bayesian

---

[4]$\alpha := 0.8 \times \min(\mathbf{y})$. See Supplementary for the details on $\alpha$

optimisation [39] and neural networks [36] surrogates are proposed for simulation-based inference [25]. In a *Bayesian coreset* context, scalable Bayesian inference [60], sparse variational inference [19, 20], active learning [68] have been proposed for large-scale dataset inference. However, all of the above only calculate posteriors, not model evidences. For evidence inference, *Annealed Importance Sampling* [64], and *Bridge Sampling* [9], Sequential Monte Carlo (SMC) [27] are popular, but *only* estimate evidence, not the posterior.

**Bayesian quadrature** The early works on BQ, which directly replaced the likelihood function with a GP [65, 66, 69], did not explicitly handle non-negative integrand constraints. Osborne et al. [67] introduced logarithmic warped GP to handle the non-negativity of the integrand, and introduced active learning for BQ, a method that selects samples based on where the variance of the integral will be minimised. Gunter et al. [37] introduced square-root GP to make the integrand function closed-form and to speed up the computation. Furthermore, they accelerated active learning by changing the objective from the variance of the integral $\mathbb{V}\mathrm{ar}[Z|\mathbf{y}]$ to simply the variance of the integrand $\mathbb{V}\mathrm{ar}[\ell(x)\pi(x)]$. Wagstaff et al. [78] introduced the first batch BQ. Chai et al. [22] generalised warped GP for BQ, and proposed probit-transformation. BQ has been extended to machine learning tasks (model selection [21], manifold learning [30], kernel learning [41]) with new acquisition function (AF) designed for each purpose. For posterior inference, VBMC [1] has pioneered that BQ can infer posterior and evidence in one go via variational inference, and [2] has extended it to the noisy likelihood case. This approach is an order of magnitude more expensive than the WSABI approach because it essentially involves BQ inside of an optimisation loop for variational inference. This paper employs WSABI-L and its AF for simplicity and faster computation. Still, our approach is a natural extension of BQ methods and compatible with the above advances. (e.g. changing the RCHQ kernel into the prospective uncertainty sampling AF for VMBC)

**Kernel quadrature** There are a number of KQ algorithms from herding/optimisation [23, 6, 48] to random sampling [5, 8]. In the context of BQ, Frank-Wolfe Bayesian quadrature (FWBQ) [13] using kernel herding has been proposed. This method proposes to do BQ with the points given by herding, but the guaranteed exponential convergence $\mathcal{O}(\exp(-cn))$ is limited to finite-dimensional RKHS, which is not the case in our setting. For general kernels, the convergence rate drops to $\mathcal{O}(1/\sqrt{n})$ [6]. Recently, a random sampling method with a good convergence rate was proposed for infinite-dimensional RKHS [44]. Based on Carathéodory's theorem for the convex hull, it efficiently calculates a reduced measure for a larger empirical measure. In this paper, we call it *random convex hull quadrature* (RCHQ) and use it in combination with the BQ methods. (See Supplementary)

# 4 Proposed Method: BASQ

## 4.1 General idea

We now introduce our algorithm, named *Bayesian alternately subsampled quadrature* (BASQ).

**Kernel recombination for batch selection** Batch WSABI [78] selects batch samples based on the AF, taking samples having the maximum AF values greedily via gradient-based optimisation with multiple starting points. The computational cost of this sampling scheme increases exponentially with the batch size and/or the problem dimension. Moreover, this method is often only locally optimal [79]. We adopt a scalable, gradient-free optimisation via KQ algorithm based on kernel recombination [44]. Surprisingly, Huszár et al. [48] pointed out the equivalence between BQ and KQ with optimised weights. KQ can select $n$ samples from the $N$ candidate samples $\mathbf{X}_{\mathrm{rec}}$ to efficiently reduce the worst-case error. The problem in batch BQ is selecting $n$ samples from the probability measure $\pi(x)$ that minimises integral variance. When subsampling $N$ candidate samples $\mathbf{X}_{\mathrm{rec}} \sim \pi(x)$, we can regard this samples $\mathbf{X}_{\mathrm{rec}}$ as an empirical measure $\pi_{\mathrm{emp}}(x)$ approximating the true probability measure $\pi(x)$ if $n \ll N$. Therefore, applying KQ to select $n$ samples that can minimise $\mathbb{V}\mathrm{ar}[\ell(x)\pi_{\mathrm{emp}}(x)]$ is equivalent to selecting $n$ batch samples for batch BQ. As more observations make surrogate model $\ell(x)$ more accurate, the empirical integrand model $\ell(x)\pi_{\mathrm{emp}}(x)$ approaches to the true integrand model $\ell_{\mathrm{true}}(x)\pi(x)$. This subsampling scheme allows us to apply any KQ methods for batch BQ. However, such a dual quadrature scheme tends to be computationally demanding. Hayakawa et al. [44] proposed an efficient algorithm based on kernel recombination, RCHQ, which automatically

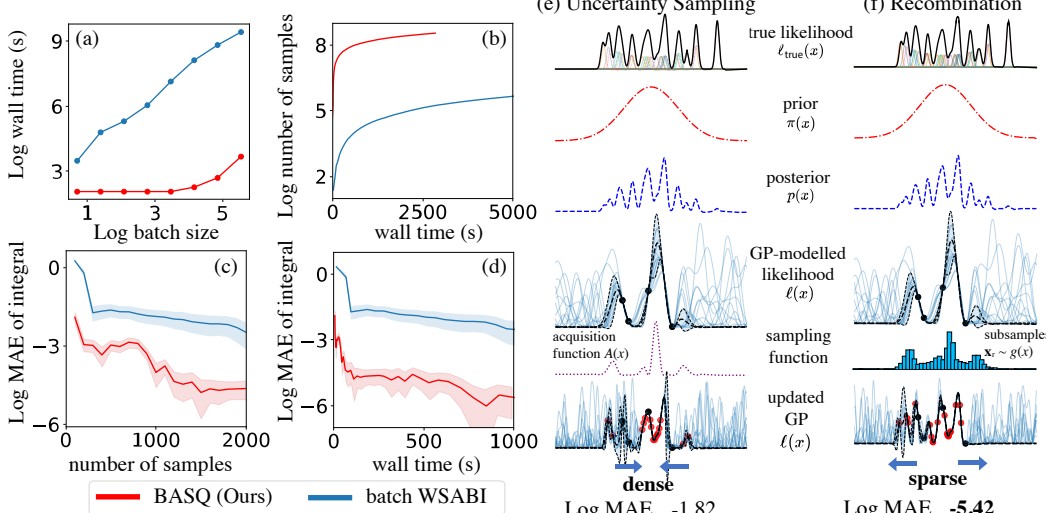

Figure 1: Performance comparison of our algorithm BASQ against batch WSABI [78]. All evaluation was performed with the likelihood of a mixture of N-dimensional Gaussians. (a), (b), (c), (d) 10-dimensional Gaussians, (e), (f) univariate Gaussians.

returns a sparse set of $n$ samples based on Carathéodory's theorem. The computational cost of batch size $n$ for our algorithm, BASQ, is lower than $\mathcal{O}(NM + M^2 \log n + Mn^2 \log(N/n))$ [44].

**Alternately subsampling**    The performance of RCHQ relies upon the quality of a predefined kernel. Thus, we add BQ elements to KQ in return; making RCHQ an online algorithm. Throughout the sequential batch update, we pass the BQ-updated kernels to RCHQ.[5] This enables RCHQ to exploit the function shape information to select the best batch samples for minimising $\mathbb{V}\mathrm{ar}[\ell(x)\pi_{\mathrm{emp}}(x)]$. This corresponds to the batch maximisation of the model evidence for $\ell(x)$. Then, BQ optimises the hyperparameters based on samples from true likelihood $\ell_{\mathrm{true}}(x)$, which corresponds to an optimised kernel preset for RCHQ in the next round. These alternate updates characterise our algorithm, BASQ. (See Supplementary)

**Importance sampling for uncertainty**    We added one more BQ element to RCHQ; *uncertainty sampling*. RCHQ relies upon the quality of subsamples from the prior distribution. However, sharp, multimodal, or high-dimensional likelihoods make it challenging to find prior subsamples that overlap over the likelihood distribution. This typically happens in Bayesian inference when the likelihood is expressed as the product of Gaussians, which gives rise to a very sharp likelihood peak. The likelihood of big data tends to become multimodal [70]. Therefore, we adopt *importance sampling*, gravitating subsamples toward the meaningful region, and correct the bias via its weights. We propose a mixture of prior and an AF as a guiding *proposal distribution*. The prior encourages global (rather than just local) optimisation, and the AF encourages sampling from uncertain areas for faster convergence. However, sampling from AF is expensive. We derive an efficient sparse Gaussian mixture sampler. Moreover, introducing square-root warping [37] enables the sampling distribution to factorise, yielding faster sampling.

**Summary of contribution**    We summarised the key differences between batch WSABI and BASQ in Figure 1.[6] (a) shows that BASQ is more scalable in batch size. (b) clarifies that BASQ can sample

---

[5]For updating kernel, the kernel to be updated is $C_{\mathbf{y}}$, not the kernel $K$. The kernel $K$ just corresponds to the *prior* belief in the distribution of $\ell$, so once we have observed the samples $\mathbf{X}$ (and $\mathbf{y}$), the variance to be minimised becomes $C_{\mathbf{y}}$.

[6]We randomly varied the number of components between 10 and 15, setting their variance uniformly at random between 1 and 4, and setting their means uniformly at random within the box bounded by [-3,3] in all dimensions. The weights of Gaussians were randomly generated from a uniform distribution, but set to be one after integration. mean absolute error (MAE) was adopted for the evaluation of integral estimation.

Table 1: BASQ algorithm

**Algorithm 1**: Bayesian Alternately Subsampled Quadrature (BASQ)

**Notation**: $x_{\text{init}}$: initial guess, $k$: a convergence criterion,
$n$: batch size, $M$: Nyström sample size, $N$: recombination sample size,
$\ell$: GP-modelled likelihood, $\ell_{\text{true}}$: true likelihood,
$\pi(x)$: prior, $p(x)$: posterior, $A(x)$: AF, $K$: kernel,
$S_{\text{nys}}, S_{\text{rec}}$: samplers for Nyström and recombination, respectively

**Input**: prior $\pi(x)$, true likelihood function $\ell_{\text{true}}$
**Output**: posterior $p(x)$, the mean and variance of model evidence $\mathbb{E}[Z \mid \mathbf{y}], \text{var}[Z \mid \mathbf{y}]$

1: $C_{\mathbf{y}}^L, A(x) \leftarrow \text{InitialiseGPs}(x_{\text{init}})$      # Initialise GPs with initial guess
2: $S_{\text{nys}}, S_{\text{rec}} \leftarrow \text{SetSampler}(A(x), \pi(x))$      # Set samplers
3: **while** $\text{var}[Z \mid \mathbf{y}] > k$:
4:      $\mathbf{X}_{\text{nys}} \sim S_{\text{nys}}(M)$      # Samples $M$ points for test function $\varphi$
5:      $(\mathbf{w}_{\text{rec}}, \mathbf{X}_{\text{rec}}) \sim S_{\text{rec}}(N)$      # Samples $N$ points for recombination
6:      $\varphi_1, ..., \varphi_{n-1} \leftarrow \text{Nyström}(\mathbf{X}_{\text{nys}}, C_{\mathbf{y}}^L)$      # Define $n-1$ test functions
7:      **solve a kernel recombination problem**
8:          Find an $n$-point subset $\mathbf{X}_{\text{quad}} \subset \mathbf{X}_{\text{rec}}$ and $\mathbf{w}_{\text{quad}} \geq \mathbf{0}$
9:          s.t. $\mathbf{w}_{\text{quad}}^\top \varphi_i(\mathbf{X}_{\text{quad}}) = \mathbf{w}_{\text{rec}}^\top \varphi_i(\mathbf{X}_{\text{rec}}), \mathbf{w}_{\text{quad}}^\top \mathbf{1} = \mathbf{w}_{\text{rec}}^\top \mathbf{1}$
10:          **return** $\mathbf{X}_{\text{quad}}$      # The sparse set of $n$ samples
11:      $\mathbf{y} = \text{Parallel}(\ell_{\text{true}}(\mathbf{X}_{\text{quad}}))$      # Parallel computing of likelihood
12:      $K \leftarrow \text{Update}(\mathbf{X}_{\text{quad}}, \mathbf{y})$      # Train GPs
13:      $C_{\mathbf{y}}^L, A(x) \leftarrow \text{OptHypersThenUpdate}(K)$      # Type II MLE optimisation
14:      $S_{\text{nys}}, S_{\text{rec}} \leftarrow \text{ResetSampler}(A(x), \pi(x))$      # Reset samplers with the updated $A(x)$
15:      $\mathbb{E}[Z \mid \mathbf{y}], \text{var}[Z \mid \mathbf{y}] \leftarrow \text{BayesQuad}(m_{\mathbf{y}}^L, C_{\mathbf{y}}^L, \pi(x))$ # Calculate via Eqs. (3) and (4)
16: **return** $p(x), \mathbb{E}[Z \mid \mathbf{y}], \text{var}[Z \mid \mathbf{y}]$

10 to 100 times as many samples in the same time budget as WSABI, supported by the efficient sampler and RCHQ. (c) states the convergence rate of BASQ is faster than WSABI, regardless of computation time. (d) demonstrates the combined acceleration in wall time. While the batch WSABI reached $10^{-1}$ after 1,000 seconds passed, BASQ was within seconds. (e) and (f), visualised how RCHQ selects sparser samples than batch WSABI. This clearly explains that gradient-free kernel recombination is better in finding the global optimal than multi-start optimisation. These results demonstrate that we were able to combine the merits of BQ and KQ (see Supplementary). We further tested with various synthetic datasets and real-world tasks in the fields of lithium-ion batteries and material science. Moreover, we mathematically analyse the convergence rate with proof in Section 6.

## 4.2 Algorithm

Table 1 illustrates the pseudo-code for our algorithm. Rows 4 - 10 correspond to RCHQ, and rows 11 - 15 correspond to BQ. We can use the variance of the integral $\mathbb{V}\text{ar}[Z|\mathbf{y}]$ as a convergence criterion. For hyperparameter optimisation, we adopt the type-II maximum likelihood estimation (MLE) to optimise hyperparameters via L-BFGS [18] for speed.

**Importance sampling for uncertainty**    Lemma 1 in the supplementary proves the optimal upper bound of the proposal distribution $g(x) \approx \sqrt{K(x,x)}f(x) = \sqrt{C_{\mathbf{y}}^L(x,x)}f(x)$, where $f(x) :=$ $\pi(x)$. However, sampling from square-root variance is intractable, so we linearised to $g(x) \approx 0.5(1 + C_{\mathbf{y}}^L(x,x))f(x)$. To correct the linearisation error, the coefficient 0.5 was changed into the hyperparameter $r$, which is defined as follows:

$$g(x) = (1-r)f(x) + r\tilde{A}(x), \quad 0 \leq r \leq 1 \tag{8}$$

$$\tilde{A}(x) = \frac{C_{\mathbf{y}}^L(x,x)\pi(x)}{\int C_{\mathbf{y}}^L(x,x)\pi(x)dx}, \tag{9}$$

$$\mathbf{w}_{\text{IS}}(x) = f(x)/g(x), \tag{10}$$

where $w_{IS}$ is the weight of the importance sampling. While $r = 1$ becomes the pure uncertainty sampling, $r = 0$ is the vanilla MC sampler.

**Efficient sampler**   Sampling from $A(x)$, a mixture of Gaussians, is expensive, as some mixture weights are negative, preventing the usual practice of weighted sampling from each Gaussian. As the warped kernel $C_{\mathbf{y}}^L(x, x)$ is also computationally expensive, we adopt a *factorisation trick*:

$$Z = \int \ell(x)\pi(x)\,\mathrm{d}x = \alpha + \frac{1}{2}\int |\tilde{\ell}(x)|^2 \pi(x)\,\mathrm{d}x \approx \alpha + \frac{1}{2}\int |\tilde{\ell}(x)| f(x)\,\mathrm{d}x, \tag{11}$$

where we have changed the distribution of interest to $f(x) = |\tilde{m}_{\mathbf{y}}(x)|\pi(x)$. This is doubly beneficial. Firstly, the distribution of interest $f(x)$ will be updated over iterations. The previous $f(x) = \pi(x)$ means the subsample distribution eventually obeys prior, which is disadvantageous if the prior does not overlap the likelihood sufficiently. On the contrary, the new $f(x)$ narrows its region via $|\tilde{m}_{\mathbf{y}}(x)|$. Secondly, the likelihood function changed to $|\tilde{\ell}(x)|$, thus the kernels shared with RCHQ changed into cheap warped kernel $\tilde{C}_{\mathbf{y}}(x, x)$. This reduces the computational cost of RCHQ, and the sampling cost of $A(x)$. Now $A(x) = \tilde{C}_{\mathbf{y}}(x)\pi(x)$, which is also a Gaussian mixture, but the number of components is significantly lower than the original AF (9). As $\tilde{C}_{\mathbf{y}}(x)$ is positive, the positive weights of the Gaussian mixture should cover the negative components. Interestingly, in many cases, the positive weights vary exponentially, which means that limited number of components dominate the functional shape. Thus, we can ignore the trivial components for sampling.[7]  Then we adopt SMC [53] to sample $A(x)$. We have a highly-calibrated proposal distribution of sparse Gaussian mixture, leading to efficient resampling from real $A(x)$ (see Supplementary).

**Variants of proposal distribution**   Although (8) has mathematical motivation, sometimes we wish to incorporate prior information not included in the above procedure. We propose two additional "biased" proposal distributions. The first case is where we know both the maximum likelihood points and the likelihood's unimodality. This is typical in Bayesian inference because we can obtain (sub-)maximum points via a maximum a posteriori probability (MAP) estimate. In this case, we know exploring around the perfect initial guess is optimal rather than unnecessarily exploring an uncertain region. Thus, we introduce the initial guess believer (IGB) proposal distribution, $g_{IGB}(x)$. This is written as $g_{IGB}(x) = (1 - r)\pi(x) + r\sum_{i=1} w_{i,IGB}\mathcal{N}(x; X_i, \mathbf{W})$, where $w_{i,IGB} = \{0 \text{ if } y_i \leq 0, \text{ else } 1\}$, $X_i \in \mathbf{X}$. This means exploring only the vicinity of the observed data $\mathbf{X}$. The second case is where we know the likelihood is multimodal. In this case, determining all peak positions is most beneficial. Thus more explorative distribution is preferred. As such, we introduce the uncertainty believer (UB) proposal distribution, $g_{UB}(x)$. This is written as $g_{UB}(x) = A(x)$, meaning pure uncertainty sampling. To contrast the above two, we term the proposal distribution in Eq. (8) as integral variance reduction (IVR) $g_{IVR}(x)$.

## 5   Experiments

Given our new model BASQ, with three variants of the proposal distribution, IVR, IGB, and UB, we now test for speed against MC samplers and batch WSABI. We compared with three NS methods [71, 29, 46, 14, 15], coded with [72, 17]. According to the review [16], MLFriends is the state-of-the-art NS sampler to date. The code is implemented based on [77, 34, 42, 76, 38, 31, 10, 7], and code around kernel recombination [24, 44] with additional modification. All experiments on synthetic datasets were averaged over 10 repeats, computed in parallel with multicore CPUs, without GPU for fair comparison.[8] The posterior distribution of NS was estimated via KDE with weighted samples [33]. For maximum speed performance, batch size was optimised for each method in each dataset, in fairness to the competitors. Batch WSABI needs to optimise batch size to balance the likelihood query cost and sampling cost, because sampling cost increases rapidly with batch size, as shown in Figure 1(a). Therefore, it has an optimal batch size for faster convergence. By wall time cost, we exclude the cost of integrand evaluation; that is, the wall time cost is the overhead cost of batch evaluation. Details can be found in the Supplementary.

---

[7]Negative elements in the matrices only exist in $K(\mathbf{X}, \mathbf{X})^{-1}$, which can be drawn from the memory of the GP regression model without additional calculation. The number of positive components is half of the matrix on average, resulting in $\mathcal{O}(n^2/2)$. Then, taking the threshold via the inverse of the recombination sample size $N$, the number of components becomes $n_{comp} \ll n^2$, resulting in sampling complexity $\mathcal{O}(n^2/2 + n_{comp}N)$.

[8]Performed on MacBook Pro 2019, 2.4 GHz 8-Core Intel Core i9, 64 GB 2667 MHz DDR4

## 5.1 Synthetic problems

We evaluate all methods on three synthetic problems. The goal is to estimate the integral and posterior of the likelihood modelled with the highly multimodal functions. Prior was set to a two-dimensional multivariate normal distribution, with a zero mean vector, and covariance whose diagonal elements are 2. The optimised batch sizes for each methods are BASQ: 100, batch WSABI: 16. The synthetic likelihood functions are cheap (0.5 ms on average). This is advantageous setting for NS: Within 10 seconds, the batch WSABI, BASQ, and NS collected 32, 600, 23225 samples, respectively. As for the metrics, posterior estimation was tested with Kullback-Leibler (KL) upon random 10,000 samples from true posterior. Evidence was evaluated with MAE, and ground truth was derived analytically.

**Likelihood functions** *Branin-Hoo* [49] is 8 modal function in two-dimensional space. *Ackley* [73] is a highly multimodal function with point symmetric periodical peaks in two-dimensional space. *Oscillatory function* [32] is a highly multimodal function with reflection symmetric periodical peaks of highly-correlated ellipsoids in two-dimensional space.

## 5.2 Real-world dataset

We consider three real-world applications with expensive likelihoods, which are simulator-based and hierarchical GP. We adopted the empirical metric due to no ground truth. For the posterior, we can calculate the true conditional posterior distribution along the line passing through ground truth parameter points. Then, evaluate the posterior with root mean squared error (RMSE) against 50 test samples for each dimension. For integral, we compare the model evidence itself. Expensive likelihoods makes the sample size per wall time amongst the methods no significant difference, whereas rejection sampling based NS dismiss more than 50% of queried samples. The batch sizes are BASQ: 32, batch WSABI: 8. (see Supplementary)

**Parameter estimation of the lithium-ion battery simulator**  : The simulator is the SPMe [61], [9] estimating 3 simulation parameters at a given time-series voltage-current signal (the diffusivity of lithium-ion on the anode and cathode, and the experimental noise variance). Prior is modified to log multivariate normal distribution from [4]. Each query takes 1.2 seconds on average.

**Parameter estimation of the phase-field model**  : The simulator is the phase-field model [61], [10] estimating 4 simulation parameters at given time-series two-dimensional morphological image (temperature, interaction parameter, Bohr magneton coefficient, and gradient energy coefficient). Prior is a log multivariate normal distribution. Each query takes 7.4 seconds on average.

**Hyperparameter marginalisation of hierarchical GP model**   The hierarchical GP model was designed for analysing the large-scale battery time-series dataset from solar off-grid system field data [3].[8] For fast estimation of parameters in each GP, the recursive technique [74] is adopted. The task is to marginalise 5 GP hyperparameters at given hyperprior, which is modified to log multivariate normal distribution from [3]. Each query takes 1.1 seconds on average.

## 5.3 Results

We find BASQ consistently delivers strong empirical performance, as shown in Figure 2. On all benchmark problems, BASQ-IVR, IGB, or UB outperform baseline methods except in the battery simulator evidence estimation. The very low-dimensional and sharp unimodal nature of this likelihood could be advantageous for biased greedy batch WSABI, as IGB superiority supports this viewpoint. This suggests that BASQ could be a generally fast Bayesian solver as far as we investigated. In the multimodal setting of the synthetic dataset, BASQ-UB outperforms, whereas IVR does in a simulator-based likelihoods. When comparing each proposal distribution, BASQ-IVR was the performant. Our results support the general use of IVR, or UB if the likelihood is known to be highly multimodal.

---

[9] SPMe code used was translated into Python from MATLAB [11, 12]. This open-source code is published under the BSD 3-clause Licence. See more information on [11]

[10] Code used was from [54, 3]. All rights of the code are reserved by the authors. Thus, we do not redistribute the original code.

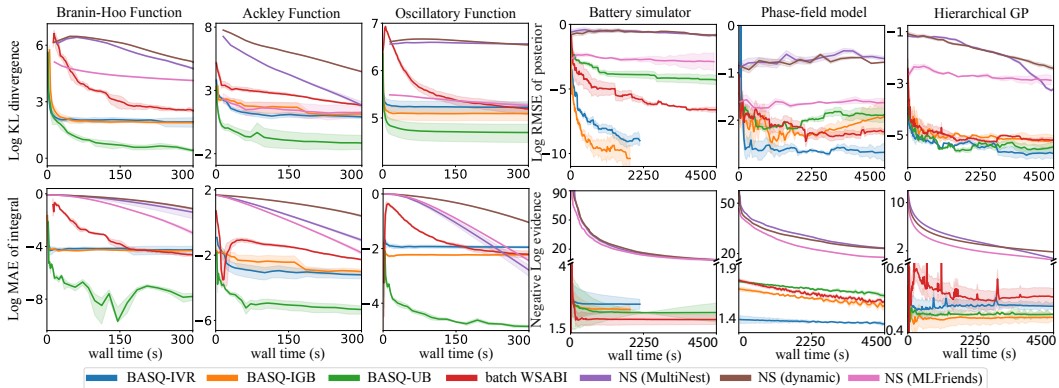

Figure 2: Time in seconds vs. KL divergence for posterior, and MAE for evidence in the synthetic datasets. Time in seconds vs. RMSE for posterior and evidence itself in the real-world dataset.

## 6 Convergence analysis

We analysed the convergence over single iteration on a simplified version of BASQ, which assumes the BQ is modelled with vanilla BQ, without batch and hyperparameter updates. Note that the kernel $K$ on $\mathbb{R}^d$ in this section refers to the given covariance kernel of GP at each step. We discuss the convergence of BASQ in one iteration. We consider the importance sampling: Let $f$ be a probability density on $\mathbb{R}^d$ and $g$ be another density such that $f = \lambda g$ with a nonnegative function $\lambda$. Let us call such a pair, $(f, g)$, a density pair with weight $\lambda$.

We approximate the kernel $K$ with $K_0 = \sum_{i=1}^{n-1} c_i \varphi_i(x)\varphi_i(y)$. In general, we can apply the kernel recombination algorithm [59, 75] with the weighted sample $(\mathbf{w}_{\mathrm{rec}}, \mathbf{X}_{\mathrm{rec}})$ to obtain a weighted point set $(\mathbf{w}_{\mathrm{quad}}, \mathbf{X}_{\mathrm{quad}})$ of size $n$ satisfying $\mathbf{w}_{\mathrm{quad}}^\top \varphi_i(\mathbf{X}_{\mathrm{quad}}) = \mathbf{w}_{\mathrm{rec}}^\top \varphi_i(\mathbf{X}_{\mathrm{rec}})$ $(i = 1, \dots, n-1)$ and $\mathbf{w}_{\mathrm{quad}}^\top \mathbf{1} = \mathbf{w}_{\mathrm{rec}}^\top \mathbf{1}$. By modifying the kernel recombination algorithm, we can require $\mathbf{w}_{\mathrm{quad}}^\top k_1^{1/2}(\mathbf{X}_{\mathrm{quad}}) \leq \mathbf{w}_{\mathrm{rec}}^\top k_1^{1/2}(\mathbf{X}_{\mathrm{rec}})$, where $k_1^{1/2}(x) := \sqrt{K(x,x) - K_0(x,x)}$ [44]. We call such $(\mathbf{w}_{\mathrm{quad}}, \mathbf{X}_{\mathrm{quad}})$ a *proper* kernel recombination of $(\mathbf{w}_{\mathrm{rec}}, \mathbf{X}_{\mathrm{rec}})$ with $K_0$.[11] We have the following guarantee (proved in Supplementary):

**Theorem 1.** *Suppose $\int \sqrt{K(x,x)} f(x)\,\mathrm{d}x < \infty$, $\ell \sim \mathcal{GP}(m, K)$, and we are given an $(n-1)$-dimensional kernel $K_0$ such that $K_1 := K - K_0$ is also a kernel. Let $(f, g)$ be a density pair with weight $\lambda$. Let $\mathbf{X}_{\mathrm{rec}}$ be an $N$-point independent sample from $g$ and $\mathbf{w}_{\mathrm{rec}} := \lambda(\mathbf{X}_{\mathrm{rec}})$. Then, if $(\mathbf{w}_{\mathrm{quad}}, \mathbf{X}_{\mathrm{quad}})$ is a proper kernel recombination of $(\mathbf{w}_{\mathrm{rec}}, \mathbf{X}_{\mathrm{rec}})$ for $K_0$, it satisfies*

$$\mathbb{E}_{\mathbf{x}_{\mathrm{rec}}}\left[\sqrt{\mathrm{var}[Z_f \mid \mathbf{x}_{\mathrm{quad}}]}\right] \leq 2\left(\int K_1(x,x) f(x)\,\mathrm{d}x\right)^{1/2} + \sqrt{\frac{C_{K,f,g}}{N}}, \qquad (12)$$

*where $Z_f := \int \ell(x) f(x)\,\mathrm{d}x$ and $C_{K,f,g} := \int K(x,x)\lambda(x)f(x)\,\mathrm{d}x - \iint K(x,y)f(x)f(y)\,\mathrm{d}x\,\mathrm{d}y$.*

The above approximation has one source of randomness which stems from sampling $N$ points $\mathbf{x}_{\mathrm{rec}}$ from $g$. One can also apply this estimate with a random kernel and thereby introduce another source of randomness. In particular, when we use the Nyström approximation for $K_0$ (that ensures $K_1$ is a kernel [44]), then one can show that $\int K_1(x,x)f(x)\,\mathrm{d}x$ can be bounded by

$$\int K_1(x,x)\,\mathrm{d}x \leq n\sigma_n + \sum_{m=n+1}^{\infty} \sigma_m + \mathcal{O}_p\left(\frac{nK_{\max}}{\sqrt{M}}\right), \qquad (13)$$

where $\sigma_n$ is the $n$-th eigenvalue of the integral operator $L^2(f) \ni h \mapsto \int K(\cdot, y)h(y)f(y)\,\mathrm{d}x$, $K_{\max} := \sup_x K(x,x)$. However, note that unlike Eq. (12), this inequality only applies with high probability due to the randomness of $K_0$; see Supplementary for details.

---

[11]Note that the inequality constraint on the diagonal value here is only needed for theoretical guarantee, and skipping it does not reduce the empirical performance [44].

If, for example, $K$ is a Gaussian kernel on $\mathbb{R}^d$ and $f$ is a Gaussian distribution, we have $\sigma_n = \mathcal{O}(\exp(-cn^{1/d}))$ for some constant $c > 0$ (see Supplementary). So in (12) we also achieve an empirically exponential rate when $N \gg C_{K,f,g}$. RCHQ works well with a moderate $M$ in practise. Note that unlike the previous analysis [50], we do not have to assume that the space is compact. [12]

## 7 Discussion

We introduced a batch BQ approach, BASQ, capable of simultaneous calculation of both model evidence and posteriors. BASQ demonstrated faster convergence (in wall-clock time) on both synthetic and real-world datasets, when compared against existing BQ approaches and state-of-the-art NS. Further, mathematical analysis shows the possibility to converge exponentially-fast under natural assumptions. As the BASQ framework is general-purpose, this can be applied to other active learning GP-based applications, such as Bayesian optimisation [52], dynamic optimisation like control [26], and probabilistic numerics like ODE solvers [45]. Although it scales to the number of data seen in large-scale GP experiments, practical BASQ usage is limited to fewer than 16 dimensions (similar to many GP-based algorithms). However, RCHQ is agnostic to the input space, allowing quadrature in manifold space. An appropriate latent variable warped GP modelling, such as GPLVM [58], could pave the way to high dimensional quadrature in future work. In addition, while WSABI modelling limits the kernel to a squared exponential kernel, RCHQ allows to adopt other kernels or priors without a bespoke modelling BQ models. (See Supplementary). As for the mathematical proof, we do not incorporate batch and hyperparameter updates, which should be addressed in future work. The generality of our theoretical guarantee with respect to kernel and distribution should be useful for extending the analysis to the whole algorithm.

## Acknowledgments and Disclosure of Funding

We thank Saad Hamid and Xingchen Wan for the insightful discussion of Bayesian quadrature, and Antti Aitio and David Howey for fruitful discussion of Bayesian inference for battery analytics and for sharing his codes on the single particle model with electrolyte dynamics, and hierarchical GPs. We would like to thank Binxin Ru, Michael Cohen, Samuel Daulton, Ondrej Bajgar, and anonymous reviewers for their helpful comments about improving the paper. Masaki Adachi was supported by the Clarendon Fund, the Oxford Kobe Scholarship, the Watanabe Foundation, the British Council Japan Association, and Toyota Motor Corporation. Satoshi Hayakawa was supported by the Clarendon Fund, the Oxford Kobe Scholarship, and the Toyota Riken Overseas Scholarship. Harald Oberhauser was supported by the DataSig Program [EP/S026347/1], the Hong Kong Innovation and Technology Commission (InnoHK Project CIMDA), and the Oxford-Man Institute. Martin Jørgensen was supported by the Carlsberg Foundation.

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
