# Supplementary: Fast Bayesian Inference with Batch Bayesian Quadrature via Kernel Recombination

## 1 Convergence analysis

### 1.1 Proof of Theorem 1

We provide the proof of the following theorem given in the main text.

**Theorem 1.** *Suppose $\int \sqrt{K(x,x)} f(x)\,\mathrm{d}x < \infty$, $\ell \sim \mathcal{GP}(m, K)$, and we are given an $(n-1)$-dimensional kernel $K_0$ such that $K_1 := K - K_0$ is also a kernel. Let $(f, g)$ be a density pair with weight $\lambda$. Let $\mathbf{x}_{\mathrm{rec}}$ be an $N$-point independent sample from $g$ and $\mathbf{w}_{\mathrm{rec}} := \lambda(\mathbf{x}_{\mathrm{rec}})$. Then, if $(\mathbf{w}_{\mathrm{quad}}, \mathbf{x}_{\mathrm{quad}})$ is a proper recombination of $(\mathbf{w}_{\mathrm{rec}}, \mathbf{x}_{\mathrm{rec}})$ for $K_0$, it satisfies*

$$\mathbb{E}_{\mathbf{x}_{\mathrm{rec}}}\left[ \sqrt{\mathrm{var}[Z_f \mid \mathbf{x}_{\mathrm{quad}}]} \right] \leq 2\left( \int K_1(x,x) f(x)\,\mathrm{d}x \right)^{1/2} + \sqrt{\frac{C_{K,f,g}}{N}} \tag{1}$$

*where $Z_f := \int \ell(x) f(x)\,\mathrm{d}x$ and $C_{K,f,g} := \int K(x,x)\lambda(x)f(x)\,\mathrm{d}x - \iint K(x,y)f(x)f(y)\,\mathrm{d}x\,\mathrm{d}y$.*

Recall $\mathcal{H}$ is the RKHS given by the kernel $K$. As the kernel satisfies $\int \sqrt{K(x,x)} f(x)\,\mathrm{d}x < \infty$, the mean embedding

$$\mu_K(f) := \int f(x) K(x, \cdot)\,\mathrm{d}x \tag{2}$$

is a well-defined element of $\mathcal{H}$. We first discuss its approximation via importance sampling.

**Lemma 1.** *Let $f$ be a probability density on $\mathbb{R}^d$ and $g$ be another density such that $f = \lambda g$ with a nonnegative function $\lambda$. Let $\mathbf{x}_{\mathrm{rec}}$ be an $N$-point independent sample from $g$ and $\mathbf{w}_{\mathrm{rec}} = \lambda(\mathbf{x}_{\mathrm{rec}})$ be the weights. If we define $\mu_r := \frac{1}{N}\mathbf{w}_{\mathrm{rec}}^\top K(\mathbf{x}_{\mathrm{rec}}, \cdot)$ then it satisfies*

$$\mathbb{E}[\|\mu_K(f) - \mu_r\|_{\mathcal{H}}^2] = \tfrac{1}{N} C_{K,f,g}$$
$$\text{where} \quad C_{K,f,g} = \int K(x,x)\lambda(x)f(x)\,\mathrm{d}x - \iint K(x,y)f(x)f(y)\,\mathrm{d}x\,\mathrm{d}y$$

*Furthermore, the choice $g(x) \propto \sqrt{K(x,x)} f(x)$ minimises $C_{K,f,g}$, if $\lambda = K(x,x)^{-1/2}$ is well-defined.*

*Proof.* Let $\mathbf{x}_{\mathrm{rec}} = (X_1, \ldots, X_N)$, so $\mu_r = \frac{1}{N}\sum_{i=1}^N \lambda(X_i) K(X_i, \cdot)$. From (2), we have

$$\|\mu_K(f) - \mu_r\|_{\mathcal{H}}^2 = \|\mu_K(f)\|_{\mathcal{H}}^2 - 2\langle \mu_K(f), \mu_r \rangle_{\mathcal{H}} + \|\mu_r\|_{\mathcal{H}}^2 \tag{4}$$

$$= \iint K(x,y) f(x) f(y)\,\mathrm{d}x\,\mathrm{d}y - \frac{2}{N}\sum_{i=1}^N \int K(x, X_i) f(x) \lambda(X_i)\,\mathrm{d}x \tag{5}$$

$$+ \frac{1}{N^2}\sum_{i,j=1}^N K(X_i, X_j) \lambda(X_i) \lambda(X_j). \tag{6}$$

We have

$$\mathbb{E}\left[ \int K(x, X_i) f(x) \lambda(X_i)\,\mathrm{d}x \right] = \iint K(x,y) f(x)\lambda(y) g(y)\,\mathrm{d}x\,\mathrm{d}y = \iint K(x,y) f(x) f(y)\,\mathrm{d}x\,\mathrm{d}y \tag{7}$$

and for $i \neq j$

$$\mathbb{E}[K(X_i, X_j)\lambda(X_i)\lambda(X_j)] = \iint K(x,y)\lambda(x)\lambda(y) g(x) g(y)\,\mathrm{d}x\,\mathrm{d}y = \iint K(x,y) f(x) f(y)\,\mathrm{d}x\,\mathrm{d}y, \tag{8}$$

so we in total have

$$\mathbb{E}[\|\mu_K(f) - \mu_r\|_{\mathcal{H}}^2] = \frac{1}{N^2} \sum_{i=1}^{N} \mathbb{E}[K(X_i, X_i)\lambda(X_i)^2] - \frac{1}{N} \iint K(x, y)f(x)f(y) \,\mathrm{d}x \,\mathrm{d}y \tag{9}$$

$$= \frac{1}{N} \left( \int K(x, x)\lambda(x)f(x) \,\mathrm{d}x - \iint K(x, y)f(x)f(y) \,\mathrm{d}x \,\mathrm{d}y \right) = \frac{C_{K,f,g}}{N}. \tag{10}$$

We next show the optimality of $g(x) \approx \sqrt{K(x, x)}f(x)$. It suffices to consider when $\int K(x, x)\lambda(x)f(x) \,\mathrm{d}x$ is minimised as the second term is independent of $g$. From the Cauchy-Schwarz, we have

$$\int K(x, x)\lambda(x)f(x) \,\mathrm{d}x = \int K(x, x)\lambda(x)f(x) \,\mathrm{d}x \int \frac{f(x)}{\lambda(x)} \,\mathrm{d}x \geq \left( \int \sqrt{K(x, x)}f(x) \,\mathrm{d}x \right)^2, \tag{11}$$

and the equality is satisfied if $g(x) = \frac{f(x)}{\lambda(x)} \propto \sqrt{K(x, x)}f(x)$. $\qquad\square$

*Proof of Theorem 1.* Let $(\mathbf{w}_{\text{quad}}, \mathbf{x}_{\text{quad}})$ be a proper recombination of $(\mathbf{w}_{\text{rec}}, \mathbf{x}_{\text{rec}})$, and let $Q_n$ be the quadrature formula given by points $\mathbf{x}_{\text{quad}}$ and weights $\frac{1}{N}\mathbf{w}_{\text{quad}}$, i.e, $Q(h) := \frac{1}{N}\mathbf{w}_{\text{quad}}^\top h(\mathbf{x}_{\text{quad}})$. We also define $\mu_n := \frac{1}{N}\mathbf{w}_{\text{quad}}^\top K(\mathbf{x}_{\text{quad}}, \cdot)$.

A well-known fact is that the worst-case error of $Q_n$ (with respect to $f$ here) $\text{wce}(Q_n) = \sup_{\|h\| \leq 1}|Q_n(h) - \int h(x)f(x) \,\mathrm{d}x|$ satisfies $\text{wce}(Q_n) = \|\mu_K(f) - \mu_n\|_{\mathcal{H}}$ for a kernel satisfying $\int \sqrt{K(x, x)}f(x) \,\mathrm{d}x < \infty$ [9, 18]. By using this and the relation between Bayesian quadrature and kernel quadrature in the main text, we have

$$\sqrt{\text{var}[Z_f \mid \mathbf{x}_{\text{quad}}]} \leq \text{wce}(Q_n) \leq \|\mu_K(f) - \mu_r\|_{\mathcal{H}} + \|\mu_r - \mu_n\|_{\mathcal{H}} \tag{12}$$

From Lemma 1 we have $\mathbb{E}[\|\mu_K(f) - \mu_r\|_{\mathcal{H}}] \leq \mathbb{E}[\|\mu_K(f) - \mu_r\|_{\mathcal{H}}^2]^{1/2} = \sqrt{C_{K,f,g}/N}$, so it now suffices to show

$$\mathbb{E}_{\mathbf{x}_{\text{rec}}}[\|\mu_r - \mu_n\|_{\mathcal{H}}] \leq 2 \left( \int K_1(x, x)f(x) \,\mathrm{d}x \right)^{1/2}. \tag{13}$$

We first have

$$\|\mu_r - \mu_n\|_{\mathcal{H}}^2 = \frac{1}{N^2} \left( \mathbf{w}_{\text{rec}}^\top K(\mathbf{x}_{\text{rec}}, \mathbf{x}_{\text{rec}})\mathbf{w}_{\text{rec}} - 2\,\mathbf{w}_{\text{rec}}^\top K(\mathbf{x}_{\text{rec}}, \mathbf{x}_{\text{quad}})\mathbf{w}_{\text{quad}} + \mathbf{w}_{\text{quad}}^\top K(\mathbf{x}_{\text{quad}}, \mathbf{x}_{\text{quad}})\mathbf{w}_{\text{quad}} \right), \tag{14}$$

and from the recombination property we also have

$$\mathbf{w}_{\text{rec}}^\top K_0(\mathbf{x}_{\text{rec}}, \mathbf{x}_{\text{rec}})\mathbf{w}_{\text{rec}} - 2\,\mathbf{w}_{\text{rec}}^\top K_0(\mathbf{x}_{\text{rec}}, \mathbf{x}_{\text{quad}})\mathbf{w}_{\text{quad}} + \mathbf{w}_{\text{quad}}^\top K_0(\mathbf{x}_{\text{quad}}, \mathbf{x}_{\text{quad}})\mathbf{w}_{\text{quad}} = 0, \tag{15}$$

which follows from the fact that $(\mathbf{w}_{\text{rec}}, \mathbf{x}_{\text{rec}})$ and $(\mathbf{w}_{\text{quad}}, \mathbf{x}_{\text{quad}})$ give the same kernel embedding for the RKHS given by $K_0$ as the latter is a recombination of the former (see e.g. [11, Eq. 14]). By subtracting, we obtain

$$\|\mu_r - \mu_n\|_{\mathcal{H}}^2 \tag{16}$$

$$= \frac{1}{N^2} \left( \mathbf{w}_{\text{rec}}^\top K_1(\mathbf{x}_{\text{rec}}, \mathbf{x}_{\text{rec}})\mathbf{w}_{\text{rec}} - 2\,\mathbf{w}_{\text{rec}}^\top K_1(\mathbf{x}_{\text{rec}}, \mathbf{x}_{\text{quad}})\mathbf{w}_{\text{quad}} + \mathbf{w}_{\text{quad}}^\top K_1(\mathbf{x}_{\text{quad}}, \mathbf{x}_{\text{quad}})\mathbf{w}_{\text{quad}} \right) \tag{17}$$

$$= \|\mu_r^{(1)} - \mu_n^{(1)}\|_{\mathcal{H}_1}^2, \tag{18}$$

where $\mathcal{H}_1$ is the RKHS given by $K_1$ and

$$\mu_r^{(1)} := \frac{1}{N}\mathbf{w}_{\text{rec}}^\top K_1(\mathbf{x}_{\text{rec}}, \cdot), \qquad \mu_n^{(1)} := \frac{1}{N}\mathbf{w}_{\text{quad}}^\top K_1(\mathbf{x}_{\text{quad}}, \cdot). \tag{19}$$

Now, by letting $k_1^{1/2}(x) := \sqrt{K(x, x)}$, we have $\|K_1(x, \cdot)\|_{\mathcal{H}_1} = k_1^{1/2}(x)$ for a point $x$. So we have

$$\|\mu_r^{(1)}\|_{\mathcal{H}_1} \leq \frac{1}{N}\mathbf{w}_{\text{rec}}^\top k_1^{1/2}(\mathbf{x}_{\text{rec}}), \qquad \|\mu_n^{(1)}\|_{\mathcal{H}_1} \leq \frac{1}{N}\mathbf{w}_{\text{quad}}^\top k_1^{1/2}(\mathbf{x}_{\text{quad}}) \leq \frac{1}{N}\mathbf{w}_{\text{rec}}^\top k_1^{1/2}(\mathbf{x}_{\text{rec}}), \tag{20}$$

where the last inequality follows from the assumption that $(\mathbf{w}_{\mathrm{quad}}, \mathbf{x}_{\mathrm{quad}})$ is a proper recombination of $(\mathbf{w}_{\mathrm{rec}}, \mathbf{x}_{\mathrm{rec}})$. Therefore, we have the estimate

$$\|\mu_r - \mu_n\|_{\mathcal{H}} = \|\mu_r^{(1)} - \mu_n^{(1)}\|_{\mathcal{H}_1} \leq \|\mu_r^{(1)}\|_{\mathcal{H}_1} + \|\mu_n^{(1)}\|_{\mathcal{H}_1} \leq \frac{2}{N} \mathbf{w}_{\mathrm{rec}}^{\top} k_1^{1/2}(\mathbf{x}_{\mathrm{rec}}). \qquad (21)$$

Finally, to prove (13), we recall that $\mathbf{x}_{\mathrm{rec}}$ is an $N$-point independent sample from $g$ and $\mathbf{w}_{\mathrm{rec}} = \lambda(\mathbf{x}_{\mathrm{rec}})$, so we obtain

$$\mathbb{E}_{\mathbf{x}_{\mathrm{rec}}}[\|\mu_r - \mu_n\|_{\mathcal{H}}] \leq 2\,\mathbb{E}_{\mathbf{x}_{\mathrm{rec}}}\left[\frac{1}{N} \mathbf{w}_{\mathrm{rec}}^{\top} k_1^{1/2}(\mathbf{x}_{\mathrm{rec}})\right] = 2 \int \lambda(x) k_1^{1/2}(x) g(x)\,\mathrm{d}x \qquad (22)$$

$$= 2 \int \sqrt{K_1(x,x)} f(x)\,\mathrm{d}x \leq 2 \left(\int K_1(x,x) f(x)\,\mathrm{d}x\right)^{1/2}, \qquad (23)$$

where we have used Cauchy–Schwarz in the last inequality. $\qquad\square$

## 1.2 Eigenvalue dacay of integral operators

Let us consider the integral operator

$$h \mapsto \int K(\cdot, y) h(y) f(y)\,\mathrm{d}y \qquad (24)$$

where $h \in L^2(f) := \{\tilde{h} \mid \text{measurable}, \|\tilde{h}\|_{L^2(f)}^2 := \int \tilde{h}(x)^2 f(x)\,\mathrm{d}x < \infty\}$, and let $\sigma_1 \geq \sigma_2 \geq \cdots \geq 0$ be eigenvalues of this operator. This sequence of eigenvalues is known to be closely related to the convergence rate of kernel quadrature [3].

For the Nyström approximation, we have the following estimate represented by the eigenvalues:

**Theorem 2** ([11]). *For a probability density function $f$ on $\mathbb{R}^d$, let $\mathbf{x}_{\mathrm{nys}}$ be an $M$-point independent sample from $f$. Let $K_0$ be the rank-$(n-1)$ approximate kernel using $\mathbf{x}_{\mathrm{nys}}$ given by Eq. (10) in the main text. Then, $K_1 := K - K_0$ satisfies*

$$\int K_1(x,x) f(x)\,\mathrm{d}x \leq n\sigma_n + \sum_{m=n+1}^{\infty} \sigma_n + \frac{2(n-1)K_{\max}}{\sqrt{M}}\left(1 + \sqrt{2\log\frac{1}{\delta}}\right) \qquad (25)$$

*with probability at least $1 - \delta$.*

This gives a theoretical guarantee for one step of our algorithm, combined with Theorem 1.

Although the sequence of eigenvalues $\sigma_n$ does not have an obvious expression when $K$ is the kernel in the middle of our algorithms BASQ, when $K$ is a multivariate Gaussian (RBF) kernel and $f$ is also a Gaussian density, we have a concrete expression of eigenvalues [8].

Indeed, if $K(x,y) = \exp(-\epsilon^2 |x-y|^2)$ and $f(x) \propto \exp(-\alpha^2 |x|^2)$, in the case $d = 1$, we have $\sigma_n = ab^n$ for some constants $a > 0$ and $0 < b < 1$ depending on $\epsilon$ and $\alpha$. Thus, for the $d$-dimensional case, we can roughly estimate that $\sigma_n \leq a^d b^{m+1}$ if $n > m^d$. So, by only using $n$, we have

$$\sigma_n \leq a^d b^{\lceil n^{1/d}\rceil} \leq a^d b^{(n^{1/d})} = a^d \exp(-cn^{1/d}), \qquad (26)$$

for $c = \log(1/b)$.

## 2 Model Analysis

### 2.1 Ablation study

#### 2.1.1 Ablation study of sampling methods

We investigated the influence of each component using 10-dimensional Gaussian mixture likelihood. The performance is evaluated by taking the mean and standard deviation of five metrics when each model gathered 1,000 observations with $n = 100$ batch size. LogMAE is the natural logarithmic MAE between the estimated integral value and true one, and the logKL is the natural logarithmic of the KL divergence between the estimated posterior and true one. Wall time is the overhead time until

Table 1: Ablation study of sampling methods

| Sampling | | Prop. dist. | | Alternate update | | Performance metric | | | | |
|---|---|---|---|---|---|---|---|---|---|---|
| Unc. sampl. | Factor. trick | Linear IVR | Optimal IVR | Kernel Update | RCHQ | logMAE | logKL | wall time (s) | logMAE per time | logKL per time |
| | | | | ✔ | | -1.8750 ±0.0435 | -8.1366 ±0.2049 | 407.42 ±10.139 | -0.0046 ±0.0002 | -0.0200 ±0.0010 |
| | ✔ | | | ✔ | ✔ | -3.3310 ±0.5265 | -9.5934 ±0.1697 | 50.065 ±8.3153 | -0.0702 ±0.0222 | -0.1976 ±0.0362 |
| ✔ | | ✔ | | ✔ | ✔ | -3.6743 ±0.0449 | -9.6108 ±0.1363 | 367.63 ±26.565 | -0.0100 ±0.0008 | -0.0263 ±0.0023 |
| ✔ | ✔ | ✔ | | | ✔ | -1.5936 ±0.0016 | -7.9967 ±0.0025 | **47.499** ±8.1966 | -0.0346 ±0.0060 | -0.1735 ±0.0300 |
| ✔ | | | ✔ | ✔ | ✔ | -3.4379 ±0.2345 | **-9.7877** **±0.4589** | 810.45 ±14.468 | -0.0042 ±0.0003 | -0.0121 ±0.0008 |
| ✔ | ✔ | ✔ | | ✔ | ✔ | **-4.0138** **±0.0078** | -9.6222 ±0.17147 | 48.75 ±8.2176 | **-0.0848** **±0.0144** | **-0.2038** **±0.0379** |

gathering 1,000 observations in seconds. LogMAE per time refers to the value that logMAE divided by the wall time. LogKL per time is the same.

Uncertainty sampling (Unc. sampl.) and factorisation trick (Factor. trick) refers to the technique explained in the section 4.2 in the main paper. Linearised IVR proposal distribution (Linear IVR) is the ones with Equations (8)-(10) in the main paper, whereas the optimal IVR is the square-root kernel IVR $g(x) = \sqrt{C_{\mathbf{y}}^L(x,x)\pi(x)}$ derived from the Lemma 1. Kernel update refers to the type-II MLE to optimise the hyperparameters. RCHQ means whether or not to adopt RCHQ, if not, it means multi-start optimisation (that is, the same with batch WSABI.)

Sampling from optimal IVR of the square root is intractable, so we adopted the SMC sampling scheme. Firstly, supersamples $\mathbf{X}_{\text{super}}$ are generated from prior $\pi(x)$, then we calculate the weights $w_i = \sqrt{C_{\mathbf{y}}^L(X_i, X_i)\pi(x_i)/\pi(x_i)} = \sqrt{C_{\mathbf{y}}^L(X_i, X_i)}$, then we normalise them via $w_i^n := \frac{w_i}{\sum_i w_i}$, where $\sum_i w_i^n = 1$. At last, we resample subsamples $\mathbf{X}_{\text{quad}}$ from the supersamples $\mathbf{X}_{\text{super}}$ with the weights $w_i$. By removing the identical samples from $\mathbf{X}_{\text{quad}}$, we can construct $\mathbf{X}_{\text{quad}}$. This removal reduces the size of subsamples to approximately 100 times smaller, so we need to supersample at least 100 times larger than the size of the subsample $\mathbf{X}_{\text{quad}}$. As the size of subsamples is already large, this SMC procedure is computationally demanding.

All components were examined by removing each. The ablation study of the sampling scheme in table 1 shows that all components are essential for reducing overhead or faster convergence. Alternate update and uncertainty sampling contribute to the fast convergence, and factorisation trick and linearised proposal distribution reduce overhead with a negligible effect on convergence.

### 2.1.2  Ablation study of BQ modelling

We investigated BQ modelling influence with the same procedure of the ablation study in the previous section. The compared models are WSABI-L, WSABI-M, Vanilla BQ (VBA), and log-warp BQ (BBQ). For the details of VBQ and BBQ modelling, see sections 3.3 and 3.4. With regard to the WSABI-M modelling, it has a disadvantageous formula in the mean posterior predictive distribution

Table 2: Ablation study of BQ modelling

| BQ modelling | | | | Performance metric | | | | |
|---|---|---|---|---|---|---|---|---|
| WSABI-L | WSABI-M | VBQ (no warp) | BBQ (log warp) | logMAE | logKL | wall time (s) | logMAE per time | logKL per time |
| ✔ | | | | -4.0138 | -9.6222 | **48.75** | **-0.0848** | **-0.2038** |
| | | | | ±0.0078 | ±0.1715 | 8.2176 | 0.0144 | 0.0379 |
| | ✔ | | | **-4.0418** | **-10.083** | 814.13 | -0.0050 | -0.0123 |
| | | | | **0.0532** | **0.2088** | ±19.813 | ±0.0002 | ±0.0006 |
| | | ✔ | | -2.5842 | 12.307 | 50.901 | 0.0534 | 0.2954 |
| | | | | ±0.9978 | ±0.0158 | ±5.3634 | ±0.0252 | ±0.0254 |
| | | | ✔ | -3.1278 | 9.0425 | 54.092 | 0.0617 | -0.2038 |
| | | | | ±1.7428 | ±0.3634 | ±5.4892 | ±0.0285 | ±0.0379 |

$m_{\mathbf{y}}^L(x)$. The acquisition function is expressed as:

$$m_{\mathbf{y}}^L(x) := \alpha + \frac{1}{2}\left(m_{\mathbf{y}}(x)^2 + C_{\mathbf{y}}(x,x)\right). \tag{27}$$

Then, the expectation of the WSABI-M is no more single term;

$$\mathbb{E}[m_{\mathbf{y}}^L(x)] := \alpha + \frac{1}{2}\mathbb{E}[m_{\mathbf{y}}(x)^2] + \frac{1}{2}\mathbb{E}[C_{\mathbf{y}}(x,x)] \tag{28}$$

As such, we cannot apply the factorisation trick for speed. Thus, we should adopt the SMC sampling scheme to sample from WSABI-M as the same procedure with the square root kernel IVR (Optimal IVR) explained in the previous section. This significantly slows down the computation with WSABI-M.

The ablation study result is shown in table 2. While WSABI-M achieves slightly better accuracy than WSABI-L, it also records the slowest computation. The WSABI-M intractable expression hinders to apply the quick sampling schemes we adopted. Vanilla BQ and BBQ (log warped BQ) shows larger errors. Therefore, the WSABI-L adoption is reasonable in this setting.

### 2.1.3 Ablation study of kernel modelling

We investigated kernel modelling influence with the same procedure of the ablation study in the previous section. The compared kernels are RBF (Radial Basis Function, as known as squared exponential, or Gaussian), Matérn32, Matérn52, Polynomial, Exponential, Rational quadratic, and exponentiated quadratic. The quadrature was performed via RCHQ with the weighted sum of the mean predictive distribution of the optimised GP. Exponential kernel marked the best accuracy in the evidence inference, whereas the KL divergence of posterior is embarrassingly erroneous. This is because all kernels examined in this section is not warped; thus, the GP-modelled likelihood is not non-negative.

## 2.2 Hyperparameter sensitivity analysis

### 2.2.1 Analysis results

The hyperparameter sensitivity analysis is performed in the same setting as the previous section that adopts a ten-dimensional Gaussian mixture. The analysis was performed with functional Analysis of Variance (ANOVA) [12]. The functional ANOVA is the statistical method to decompose the variance $V$ of a black box function $f$ into additive components $VU$ associated with each subset of hyperparameters. [12] adopts random forest for efficient decomposition and marginal prediction

Table 3: Ablation study of kernel modelling

| Kenel | logMAE | logKL | wall time (s) | logMAE per time | logKL per time |
|---|---|---|---|---|---|
| RBF | -2.9598 | 12.321 | 54.730 | -0.0543 | 0.2349 |
| | ± 0.3679 | ± 0.0288 | ± 1.2979 | ± 0.0080 | ± 0.0048 |
| Matérn32 | -3.7328 | 12.312 | 53.502 | -0.0701 | 0.2355 |
| | ± 0.9608 | ± 0.0577 | ± 0.8661 | ± 0.0191 | ± 0.0026 |
| Matérn52 | -4.3773 | 12.332 | 53.925 | -0.0817 | 0.2341 |
| | ± 1.4593 | ± 0.0765 | ± 0.9662 | ± 0.0285 | ± 0.0027 |
| Polynomial | -3.7828 | 12.208 | 52.281 | -0.0728 | 0.2449 |
| | ± 0.6803 | ± 0.0673 | ± 1.5491 | ± 0.0152 | ± 0.0056 |
| Exponential | **-4.6094** | 12.268 | **54.891** | **-0.0841** | 0.2252 |
| | **± 1.3440** | ± 0.0291 | **± 0.3428** | **± 0.0250** | ± 0.0009 |
| Rational | -3.2004 | 12.309 | 62.645 | -0.0514 | **0.2059** |
| quadratic | ± 0.6515 | ± 0.0596 | ± 1.7786 | ± 0.0119 | **± 0.0046** |
| Exponentiated | -3.3361 | **11.046** | 53.306 | -0.0627 | 0.2072 |
| quadratic | ± 1.4484 | **± 0.4465** | ± 0.1992 | ± 0.0274 | ± 0.0076 |

Table 4: Sensitivity analysis with functional ANOVA

| hyperparameters | logMAE | logKL | wall time | logMAE per time | logKL per time |
|---|---|---|---|---|---|
| N | 0.0835 | 0.0465 | **0.6347** | 0.1729 | 0.1811 |
| M | 0.1041 | 0.0824 | 0.2497 | **0.6401** | **0.6789** |
| r | 0.0041 | 0.0226 | 0.0071 | 0.0040 | 0.0024 |
| N,M | 0.0710 | 0.1134 | 0.0903 | 0.1077 | 0.1206 |
| N,r | 0.1021 | 0.1299 | 0.0058 | 0.0062 | 0.0024 |
| M,r | **0.4598** | **0.4381** | 0.0059 | 0.0604 | 0.0123 |
| N,M,r | 0.1754 | 0.3116 | 0.1671 | 0.0065 | 0.0023 |

over each hyperparameter. The hyperparameters to be analysed are the number of subsamples for recombination $N$, the number of samples for the Nyström method, and the partition ratio in the IVR proposal distribution $r$. They need to satisfy the following relationship; $N \gg M > n$, where $n$ is the batch size. We typically take $n = 100$, so $M$ should be larger than at least 200, and $N$ should be larger than at least 20,000. Grid search was adopted for the hyperparameter space, with the range of $N$ = 20,000, 50,000, 100,000, 500,000, 1,000,000, $M$ = 200, 500, 1,000, 5,000, 10,000, and $r$ = 0.0, 0.25, 0.5, 0.75, 1.0, resulting in 125 data points. To compensate for the dynamic range difference, a natural logarithmic $\log N$ and $\log M$ were used as the input.

The result is shown in Table 4. Each value refers to the fraction of the decomposed variance, corresponding to the importance of each hyperparameter over the performance metric. Functional ANOVA evaluates the main effect and the pairwise interaction effect. Figure 1 shows the marginal predictions on the important two hyperparameters for each performance metric.

As the most obvious case, we will look into the wall time. The most important hyperparameter was $N$, and the second was $M$. This is well correlated to the theoretical aspect. The overhead computation time of BASQ can be decomposed into two components; subsampling and RCHQ. The $N$ subsampling is dependent on $N$ as $\mathcal{O}(n^2/2 + n_{\text{comp}}N)$. $n_{\text{comp}}$ is way less than $M$ or $n^2$ and is insensitive to the hyperparameter variation or GP updates as we designed it to be sparse. The SMC

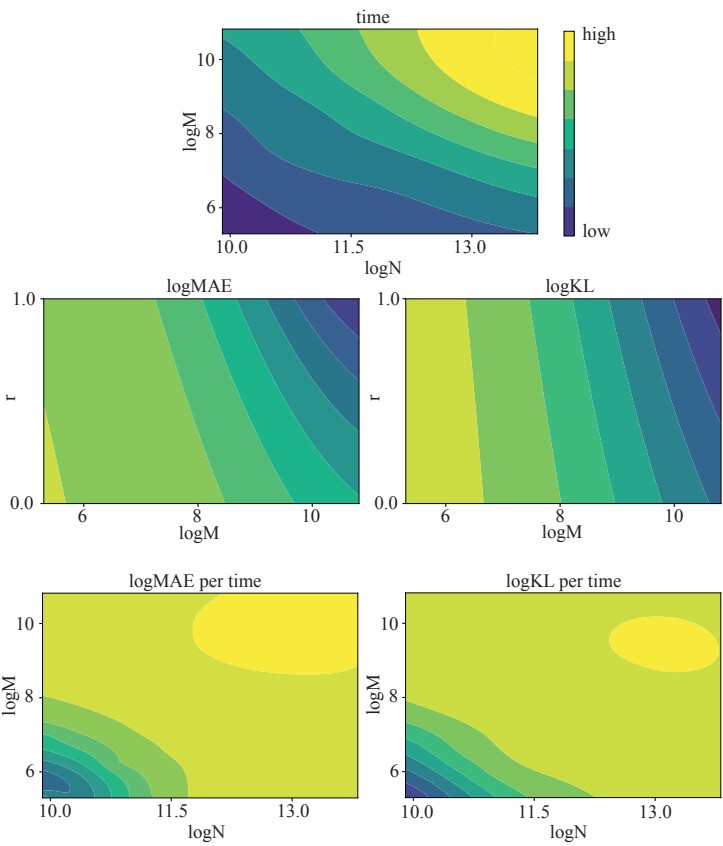

Figure 1: Sensitivity analysis of the hyperparameters over the metrics

sampling for $M$ is negligible as $\mathcal{O}(N + M)$. The RCHQ is $\mathcal{O}(NM + M^2 \log n + Mn^2 \log(N/n))$. Comparing the complexity, the RCHQ stands out. Therefore, the whole BASQ algorithm complexity is dominated by RCHQ. While $M$ has the squared component, $N$ itself is as large as $M^2$. Therefore, the selected two hyperparameters $N$ and $M$ align with the theory. Figure 1 agrees the above analysis.

The logMAE and logKL metrics have a similar trend to each other. In fact, their correlation coefficient was 0.6494. This makes sense because both metrics are determined by the functional approximation accuracy of likelihood $\ell(x)$. While increasing $M$ is always beneficial in any $r$, varying $r$ is effective in larger $M$. At last, the metrics of performance per wall time is the combination of these effects. Obviously, time is the dominant factor, so we should limit the $N$ and $M$ as small as possible. The most important hyperparameter was $M$. This is a natural consequence because $M$ affects the overhead increment less than $N$ but contributes to reducing errors more.

### 2.2.2 A guideline to select hyperparameters

The main takeaway from the functional ANOVA analysis is that the accuracy with and without the time constraint has an opposite trend. Therefore, the best hyperparameter set is dependent on the overhead time allowance. The expensiveness of likelihood evaluation determines this. Per the likelihood query time per iteration, we should increase $M$ and $N$ for faster convergence.

In the cheap likelihood case, the most relevant metric is logMAE per time and logKL per time. As we should minimise the time, we choose the minimal size for $N$ and $M$ to minimise the overhead. As the typical hyperparameter set is $(n, N, M, r) = (100, 200, 20, 000, 0.5)$, and these are the minimum values for $N$ and $M$ at given $n$. The remained choice is the selection of $r$. As shown in Figure 1, $r = 1$, namely, UB proposal distribution, was the best selection in the Gaussian mixture likelihood case. A similar trend is observed in the synthetic dataset. However, as we observed in the real-world

dataset cases, some likelihoods showed that $r = 0.5$, namely IVR proposal distribution outperformed the UB. Therefore, the $r = 1$ might be the first choice., and $r = 0.5$ is the second choice.

In the expensive likelihood case, we can increase the $N$ and $M$ because the overhead time is less significant than the likelihood query time. The logMAE and logKL without time constraints are good guidelines for tuning the hyperparameters. As $M$ is the most significant hyperparameter, we wish to increase $M$ first. However, we have to increase $N$ under the constraint $N \gg M$, necessary for RCHQ fast convergence. Empirically, we recommend increasing the $M$ three times larger than $N$ from the minimum set because the importance factor ratio in Table 4 is roughly three times. And the increment of $M$ should be corresponded to the likelihood query time $t_{\text{likelihood}}$ over the RCHQ computation time $t_{\text{RCHQ}}$. We increase the $M$ in accordance with the ratio $r_{\text{comp}} := t_{\text{likelihood}}/t_{\text{RCHQ}}$. Thus, the $M$ and $N$ is determined as $M = r_{\text{comp}} M_{\min}$, $N = \frac{r_{\text{comp}}}{3} N_{\min}$, where $M_{\min} = 200, N_{\min} = 20,000$.

## 3 Analytical form of integrals

### 3.1 Gaussian identities

$$\mathcal{N}(\mathbf{x}; \mathbf{m}_1, \boldsymbol{\Sigma}_1)\mathcal{N}(\mathbf{x}; \mathbf{m}_2, \boldsymbol{\Sigma}_2) = C_c \mathcal{N}(\mathbf{x}; \mathbf{m}_c, \boldsymbol{\Sigma}_c) \tag{29}$$

$$\int \mathcal{N}(\mathbf{x}; \mathbf{m}_1, \boldsymbol{\Sigma}_1)\mathcal{N}(\mathbf{x}; \mathbf{m}_2, \boldsymbol{\Sigma}_2)d\mathbf{x} = C_c \tag{30}$$

$$\mathcal{N}(\mathbf{A}\mathbf{x}+\mathbf{b}; \mathbf{m}, \boldsymbol{\Sigma}) = \sqrt{\frac{|2\pi(\mathbf{A}^\top \boldsymbol{\Sigma}^{-1}\mathbf{A})^{-1}|}{|2\pi\boldsymbol{\Sigma}|}} \mathcal{N}\left(\mathbf{x}; \mathbf{A}^{-1}\mathbf{m} - \mathbf{b}, (\mathbf{A}^\top \boldsymbol{\Sigma}^{-1}\mathbf{A})^{-1}\right) \tag{31}$$

where

$$\boldsymbol{\Sigma}_c = \left[\boldsymbol{\Sigma}_1^{-1} + \boldsymbol{\Sigma}_2^{-1}\right]^{-1} \tag{32}$$

$$\mathbf{m}_c = \left[\boldsymbol{\Sigma}_1^{-1} + \boldsymbol{\Sigma}_2^{-1}\right]^{-1}\left(\boldsymbol{\Sigma}_1^{-1}\mathbf{m}_1 + \boldsymbol{\Sigma}_2^{-1}\mathbf{m}_2\right) \tag{33}$$

$$C_c = \mathcal{N}(\mathbf{m}_1; \mathbf{m}_2, \boldsymbol{\Sigma}_1 + \boldsymbol{\Sigma}_2) \tag{34}$$

In the finite number of product case:

$$\prod_{i=1}^{n} \mathcal{N}(\mathbf{x}; \mathbf{m}_i, \boldsymbol{\Sigma}_i) = C_m \mathcal{N}(\mathbf{x}; \mathbf{m}_m, \boldsymbol{\Sigma}_m) \tag{35}$$

$$\int \prod_{i=1}^{n} \mathcal{N}(\mathbf{x}; \mathbf{m}_i, \boldsymbol{\Sigma}_i)d\mathbf{x} = C_m \int \mathcal{N}(\mathbf{x}; \mathbf{m}_m, \boldsymbol{\Sigma}_m)d\mathbf{x} \tag{36}$$

$$= C_m \tag{37}$$

where

$$\boldsymbol{\Sigma}_m = \left[\sum_{i=1}^{n}\left(\boldsymbol{\Sigma}_i^{-1}\right)\right]^{-1} \tag{38}$$

$$\mathbf{m}_m = \left[\sum_{i=1}^{n}\left(\boldsymbol{\Sigma}_i^{-1}\right)\right]^{-1}\sum_{i=1}^{n}\left(\boldsymbol{\Sigma}_i^{-1}\mathbf{m}_i\right) \tag{39}$$

$$C_m = \exp\left[-\frac{1}{2}\left\{(n-1)d\log 2\pi - \sum_{i=1}^{n}\log\left|\boldsymbol{\Sigma}_i^{-1}\right| + \log\left|\sum_{i=1}^{n}\left(\boldsymbol{\Sigma}_i^{-1}\right)\right|\right.\right. \tag{40}$$

$$\left.\left. + \sum_{i=1}^{n}\left((\boldsymbol{\Sigma}_i^{-1}\mathbf{m}_i)^\top \boldsymbol{\Sigma}_i(\boldsymbol{\Sigma}_i^{-1}\mathbf{m}_i)\right) - \left(\sum_{i=1}^{n}\left(\boldsymbol{\Sigma}_i^{-1}\mathbf{m}_i\right)\right)^\top \left(\sum_{i=1}^{n}\left(\boldsymbol{\Sigma}_i^{-1}\right)\right)^{-1}\left(\sum_{i=1}^{n}\left(\boldsymbol{\Sigma}_i^{-1}\mathbf{m}_i\right)\right)\right\}\right] \tag{41}$$

## 3.2 Definitions

$x_*$: predictive data points, $x_* \in \mathbb{R}^d$
$\mathbf{X}$: the observed data points, $\mathbf{X} \in \mathbb{R}^{n \times d}$
$\mathbf{y} = \ell_{\text{true}}(\mathbf{X})$: the observed likelihood, $\mathbf{y} \in \mathbb{R}^n$
$\pi(x_*) = \mathcal{N}(x_*; \mu_\pi, \boldsymbol{\Sigma}_\pi)$: prior distribution,
$v'$: a kernel variance,
$l$: a kernel lengthscale,
$K(x_*, \mathbf{X}) = v\mathcal{N}(x_*; \mathbf{X}, \mathbf{W})$: a RBF kernel,
$v = v'\sqrt{|2\pi\mathbf{W}|}$: a normalised kernel variance,
$\mathbf{W}$: a diagonal covariance matrix whose diagonal elements are the lengthscales of each dimension,
$\mathbf{W} = \begin{bmatrix} l^2 & \mathbf{0} \\ \mathbf{0} & l^2 \end{bmatrix}$
$\mathbf{I}$: The identity matrix,
$\mathbf{K}_{XX} = K(\mathbf{X}, \mathbf{X})$: a kernel over the observed data points.

## 3.3 Warped GPs as Gaussian Mixture

### 3.3.1 Mean

$$m_{\mathbf{y}}^L(x_*) = \alpha + \frac{1}{2}\tilde{m}_{\mathbf{y}}(x_*)^2 \tag{42}$$

$$= \alpha + \frac{1}{2}\mathbf{y}^T\mathbf{K}_{XX}^{-1}K(\mathbf{X}, x_*)K(x_*, \mathbf{X})\mathbf{K}_{XX}^{-1}\mathbf{y} \tag{43}$$

$$= \alpha + \frac{1}{2}\sum_{i,j}\boldsymbol{\omega}_i\boldsymbol{\omega}_j K(X_i, x_*)K(x_*, X_j) \tag{44}$$

$$= \alpha + \sum_{i,j} w_{ij}^m \mathcal{N}\left(x_*; \frac{X_i + X_j}{2}, \frac{\mathbf{W}}{2}\right) \tag{45}$$

where
Woodbury vector: $\quad \boldsymbol{\omega} = \mathbf{K}_{XX}^{-1}\mathbf{y}$,
mean weight: $\quad w_{ij}^m = \frac{1}{2}v^2\boldsymbol{\omega}_i\boldsymbol{\omega}_j\mathcal{N}(X_i; X_j, 2\mathbf{W})$

### 3.3.2 Variance

$$
\begin{aligned}
C_{\mathbf{y}}^L(x_*, x_*') &= \tilde{m}_{\mathbf{y}}(x)\tilde{C}_{\mathbf{y}}(x, x')\tilde{m}_{\mathbf{y}}(x') \\
&= [K(x_*, \mathbf{X})\boldsymbol{\omega}]^\top \left[K(x_*, x_*') - K(x_*, \mathbf{X})\mathbf{K}_{XX}^{-1}K(\mathbf{X}, x_*')\right]\left[K(x_*', \mathbf{X})\boldsymbol{\omega}\right] \\
&= \boldsymbol{\omega}^\top K(\mathbf{X}, x_*)K(x_*, x_*')K(x_*', \mathbf{X})\boldsymbol{\omega} \\
&\quad - \boldsymbol{\omega}^\top K(\mathbf{X}, x_*)K(x_*, \mathbf{X})\mathbf{K}_{XX}^{-1}K(\mathbf{X}, x_*')K(x_*', \mathbf{X})\boldsymbol{\omega} \\
&= \sum_{i,j}\boldsymbol{\omega}_i\boldsymbol{\omega}_j K(X_i, x_*)K(x_*, x_*')K(x_*', X_j) \\
&\quad - \sum_{i,j}\boldsymbol{\omega}_i\boldsymbol{\omega}_j \sum_{k,l}\boldsymbol{\Omega}_{kl}K(X_j, x_*)K(x_*, X_i)K(X_k, x_*')K(x_*', X_l) \\
&= \sum_{i,j} w_{ij}^v \mathfrak{C}_{\mathfrak{v}}(i, j) - \sum_{i,j}\sum_{k,l} w_{ijkl}^{vv}\mathfrak{C}_{\mathfrak{vv}}(i, j, k, l)
\end{aligned}
\tag{46}
$$

where
The inverse kernel weight: $\quad \boldsymbol{\Omega}_{kl} = \mathbf{K}_{XX}^{-1}(k, l)$
the first variance weight: $\quad w_{ij}^v = v^3\boldsymbol{\omega}_i\boldsymbol{\omega}_j$
the second variance weight: $\quad w_{ijkl}^{vv} = v^4\boldsymbol{\omega}_i\boldsymbol{\omega}_j\boldsymbol{\Omega}_{kl}$
the first variance Gaussian variable

$$\mathfrak{C}_{\mathfrak{v}}(i,j) = \mathcal{N}\left(\begin{bmatrix} x_* \\ x'_* \\ x'_* \end{bmatrix}; \begin{bmatrix} X_i \\ X_j \\ x_* \end{bmatrix}, \begin{bmatrix} \mathbf{W} & \mathbf{0} & \mathbf{0} \\ \mathbf{0} & \mathbf{W} & \mathbf{0} \\ \mathbf{0} & \mathbf{0} & \mathbf{W} \end{bmatrix}\right)$$

the second variance Gaussian variable

$$\mathfrak{C}_{\mathfrak{vv}}(i,j,k,l) = \mathcal{N}\left(\begin{bmatrix} x_* \\ x_* \\ x'_* \\ x'_* \end{bmatrix}; \begin{bmatrix} X_i \\ X_j \\ X_k \\ X_l \end{bmatrix}, \begin{bmatrix} \mathbf{W} & \mathbf{0} & \mathbf{0} & \mathbf{0} \\ \mathbf{0} & \mathbf{W} & \mathbf{0} & \mathbf{0} \\ \mathbf{0} & \mathbf{0} & \mathbf{W} & \mathbf{0} \\ \mathbf{0} & \mathbf{0} & \mathbf{0} & \mathbf{W} \end{bmatrix}\right)$$

$$\mathfrak{C}_{\mathfrak{v}}(i,j) = \mathcal{N}\left(\begin{bmatrix} x_* \\ x'_* \\ x'_* \end{bmatrix}; \begin{bmatrix} X_i \\ X_j \\ x_* \end{bmatrix}, \begin{bmatrix} \mathbf{W} & \mathbf{0} & \mathbf{0} \\ \mathbf{0} & \end{bmatrix}\right)$$

### 3.3.3 Symmetric variance

Here we consider $x_* = x'_*$ and $x_*$ is a sample with dimension $d$:

$$
\begin{aligned}
C_{\mathbf{y}}^L(x_*, x_*) &= C_{\mathbf{y}}^L(x_*) \\
&= \boldsymbol{\omega}^\top K(\mathbf{X}, x_*) K(x_*, x_*) K(x_*, \mathbf{X}) \boldsymbol{\omega} \\
&\quad - \boldsymbol{\omega}^\top K(\mathbf{X}, x_*) K(x_*, \mathbf{X}) \mathbf{K}_{XX}^{-1} K(\mathbf{X}, x_*) K(x_*, \mathbf{X}) \boldsymbol{\omega} \\
&= \sum_{i,j} \boldsymbol{\omega}_i \boldsymbol{\omega}_j K(X_i, x_*) K(x_*, x_*) K(x_*, X_j) \\
&\quad - \sum_{i,j} \boldsymbol{\omega}_i \boldsymbol{\omega}_j \sum_{k,l} \boldsymbol{\Omega}_{kl} K(X_j, x_*) K(x_*, X_i) K(X_k, x_*) K(x_*, X_l) \\
&= \frac{v}{\sqrt{|2\pi\mathbf{W}|}} \sum_{i,j} \boldsymbol{\omega}_i \boldsymbol{\omega}_j K(X_i, x_*) K(x_*, X_j) \\
&\quad - \sum_{i,j} \boldsymbol{\omega}_i \boldsymbol{\omega}_j \sum_{k,l} \boldsymbol{\Omega}_{kl} K(X_j, x_*) K(x_*, X_i) K(X_k, x_*) K(x_*, X_l) \\
&= \frac{v^3}{\sqrt{|2\pi\mathbf{W}|}} \sum_{i,j} \boldsymbol{\omega}_i \boldsymbol{\omega}_j \mathcal{N}(X_i; X_j, 2\mathbf{W}) \mathcal{N}\left(x_*; \frac{X_i + X_j}{2}, \frac{\mathbf{W}}{2}\right) \\
&\quad - v^4 \sum_{i,j} \left[ \boldsymbol{\omega}_i \boldsymbol{\omega}_j \sum_{k,l} \left\{ \boldsymbol{\Omega}_{kl} \mathcal{N}(X_l; X_i, 2\mathbf{W}) \mathcal{N}\left(x_*; \frac{X_l + X_i}{2}, \frac{\mathbf{W}}{2}\right) \right. \right. \\
&\qquad\qquad \left. \left. \cdot \mathcal{N}(X_k; X_j, 2\mathbf{W}) \mathcal{N}\left(x_*; \frac{X_k + X_j}{2}, \frac{\mathbf{W}}{2}\right) \right\} \right] \\
&= \sum_{i,j} w_{ij}^{'v} \mathcal{N}\left(x_*; \frac{X_i + X_j}{2}, \frac{\mathbf{W}}{2}\right) - \sum_{i,j} \sum_{k,l} w_{ijkl}^{'vv} \mathcal{N}\left(x_*; \frac{X_i + X_j + X_k + X_l}{4}, \frac{\mathbf{W}}{4}\right)
\end{aligned}
\tag{47}
$$

where
$$w_{ij}^{'v} = \frac{h^3}{\sqrt{|2\pi\mathbf{W}|}} \boldsymbol{\omega}_i \boldsymbol{\omega}_j \mathcal{N}(X_i; X_j, 2\mathbf{W})$$
$$w_{ijkl}^{'vv} = h^4 \boldsymbol{\omega}_i \boldsymbol{\omega}_j \boldsymbol{\Omega}_{kl} \mathcal{N}(X_l; X_i, 2\mathbf{W}) \mathcal{N}(X_k; X_j, 2\mathbf{W}) \mathcal{N}\left(\frac{X_k + X_j}{2}; \frac{X_i + X_l}{2}, \mathbf{W}\right)$$

### 3.4 Model evidence

The distribution over integral $Z$ is given by:

$$
p(Z|\mathbf{y}) = \int p\big(Z|\ell(x_*)\big) p\big(\ell(x_*)|\mathbf{y}\big) dx \tag{48}
$$

$$
= p\big(Z|\ell(x_*)\big) \mathcal{N}\big(\ell(x_*); m_{\mathbf{y}}^L(x_*), C_{\mathbf{y}}^L(x_*)\big) \tag{49}
$$

$$
= \mathcal{N}\big(Z; \mathbb{E}[Z|\mathbf{y}], \mathrm{var}[Z|\mathbf{y}]\big) \tag{50}
$$

### 3.4.1 Mean of the integral

$$\mathbb{E}\big[Z|\mathbf{y}\big] = \mathbb{E}\big[m_{\mathbf{y}}^L\big] \tag{51}$$

$$= \int m_{\mathbf{y}}^L(x_*)\pi(x_*)dx_* \tag{52}$$

$$= \alpha + \frac{1}{2}\int \tilde{m}_{\mathbf{y}}^2(x_*)\pi(x_*)dx_* \tag{53}$$

$$= \alpha + \sum_{i,j} w_{ij}^m \int \mathcal{N}\left(x_*; \frac{X_i + X_j}{2}, \frac{\mathbf{W}}{2}\right)\mathcal{N}(x_*; \mu_\pi, \boldsymbol{\Sigma}_\pi)dx_* \tag{54}$$

$$= \alpha + \sum_{i,j} w_{ij}^m \mathcal{N}\left(\frac{X_i + X_j}{2}; \mu_\pi, \frac{\mathbf{W}}{2} + \boldsymbol{\Sigma}_\pi\right) \tag{55}$$

### 3.4.2 Variance of the integral

$$\mathrm{var}\big[Z|\mathbf{y}\big] = \mathrm{var}\big[C_{\mathbf{y}}^L\big]$$

$$= \iint \pi(x_*)C_{\mathbf{y}}^L(x_*, x_*^{'})\pi(x_*^{'})dx_* dx_*^{'}$$

$$= \iint \Big(\boldsymbol{\omega}^\top K(\mathbf{X}, x_*)K(x_*, x_*^{'})K(x_*^{'}, \mathbf{X})\boldsymbol{\omega}\pi(x_*)\pi(x_*^{'})$$

$$\quad -\boldsymbol{\omega}^\top K(\mathbf{X}, x_*)K(x_*, \mathbf{X})\mathbf{K}_{XX}^{-1}K(\mathbf{X}, x_*^{'})K(x_*^{'}, \mathbf{X})\boldsymbol{\omega}\pi(x_*)\pi(x_*^{'})\Big)dx_* dx_*^{'}$$

$$= \sum_{i,j}\boldsymbol{\omega}_i\boldsymbol{\omega}_j \iint K(X_i, x_*)K(x_*, x_*^{'})K(x_*^{'}, X_j)\pi(x_*)\pi(x_*^{'})dx_* dx_*^{'}$$

$$\quad - \sum_{i,j}\boldsymbol{\omega}_i\boldsymbol{\omega}_j \sum_{k,l}\boldsymbol{\Omega}_{kl}\iint K(X_j, x_*)K(x_*, X_i)K(X_k, x_*^{'})K(x_*^{'}, X_l)\pi(x_*)\pi(x_*^{'})dx_* dx_*^{'}$$

$$= \sum_{i,j}\boldsymbol{\omega}_i\boldsymbol{\omega}_j h^3 \int\left[\mathcal{N}(x_*^{'}; \mu_\pi, \Sigma_\pi)\mathcal{N}(x_*^{'}; X_j, \mathbf{W})\int \mathcal{N}(x_*; X_i, \mathbf{W})\mathcal{N}(x_*; x_*^{'}, \mathbf{W})\mathcal{N}(x_*; \mu_\pi, \Sigma_\pi)dx_*\right]dx_*^{'}$$

$$\quad - \sum_{i,j}\sum k,l\,\boldsymbol{\omega}_i\boldsymbol{\omega}_j\boldsymbol{\Omega}_{kl}h^4\int \mathcal{N}(x_*; X_j, \mathbf{W})\mathcal{N}(x_*; X_i, \mathbf{W})\mathcal{N}(x_*; \mu_\pi, \Sigma_\pi)dx_*$$

$$\quad\quad\quad \cdot \int \mathcal{N}(x_*^{'}; X_k, \mathbf{W})\mathcal{N}(x_*^{'}; x_l, \mathbf{W})\mathcal{N}(x_*^{'}; \mu_\pi, \Sigma_\pi)dx_*^{'}$$

$$= \sum_{i,j}\boldsymbol{\omega}_i\boldsymbol{\omega}_j h^3\int \mathcal{N}(x_*^{'}; \mu_\pi, \Sigma_\pi)\mathcal{N}(x_*^{'}; X_j, \mathbf{W})\mathcal{N}(x_*^{'}; X_i, \mathbf{W})\mathcal{N}\left(\frac{X_i + x_*^{'}}{2}; \mu_\pi, \frac{\mathbf{W}}{2} + \boldsymbol{\Sigma}_\pi\right)dx_*^{'}$$

$$\quad - \sum_{i,j}\sum k,l\,\boldsymbol{\omega}_i\boldsymbol{\omega}_j\boldsymbol{\Omega}_{kl}h^4\left[\mathcal{N}(X_j; X_i, 2W)\mathcal{N}\left(\frac{X_i + X_j}{2}; \mu_\pi, \frac{\mathbf{W}}{2} + \boldsymbol{\Sigma}_\pi\right)\right]$$

$$\quad\quad\quad \cdot\left[\mathcal{N}(X_k; x_l, 2W)\mathcal{N}\left(\frac{X_k + X_l}{2}; \mu_\pi, \frac{\mathbf{W}}{2} + \boldsymbol{\Sigma}_\pi\right)\right]$$

$$= \sum_{i,j}\boldsymbol{\omega}_i\boldsymbol{\omega}_j h^3\int \mathcal{N}(x_*^{'}; \mu_\pi, \boldsymbol{\Sigma}_\pi)\mathcal{N}(x_*^{'}; X_i, \mathbf{W})\mathcal{N}(x_*^{'}; X_j, \mathbf{W})2^d\mathcal{N}\left(x_*^{'}; 2\mu_\pi - X_i/2, 2\mathbf{W} + 4\boldsymbol{\Sigma}_\pi\right)dx_*^{'}$$

$$\quad - \sum_{i,j}\sum_{k,l}w_{ijkl}^{vv}\mathfrak{K}_{vv}(i, j, k, l)$$

$$= \sum_{i,j}2^d w_{ij}^v\mathfrak{K}_v(i, j) - \sum_{i,j}\sum_{k,l}w_{ijkl}^{vv}\mathfrak{K}_{vv}(i, j, k, l)$$

$$\tag{56}$$

where

$$\mathfrak{K}_{\mathfrak{v}}(i,j) = \mathcal{N}(X_i; X_j, 2\mathbf{W})\mathcal{N}\left(\frac{X_i + X_j}{2}; \mu_\pi, \frac{\mathbf{W}}{2} + \boldsymbol{\Sigma}_\pi\right) \tag{57}$$

$$\mathcal{N}\left[\left(2\mathbf{W}^{-1} + \boldsymbol{\Sigma}_\pi^{-1}\right)^{-1}\left(\mathbf{W}^{-1}(X_i + X_j) + \boldsymbol{\Sigma}_\pi^{-1}\mu_\pi\right); 2\mu_\pi - \frac{X_i}{2}, \left(2\mathbf{W}^{-1} + \boldsymbol{\Sigma}_\pi^{-1}\right)^{-1} + 2\mathbf{W} + 2\boldsymbol{\Sigma}_\pi\right] \tag{58}$$

$$\mathfrak{K}_{\mathfrak{v}\mathfrak{v}}(i,j,k,l) = \left[\mathcal{N}(X_j; X_i, 2\mathbf{W})\mathcal{N}\left(\frac{X_i + X_j}{2}; \mu_\pi, \frac{\mathbf{W}}{2} + \boldsymbol{\Sigma}_\pi\right)\right]\left[\mathcal{N}(X_k; X_l, 2\mathbf{W})\mathcal{N}\left(\frac{X_k + X_l}{2}; \mu_\pi, \frac{\mathbf{W}}{2} + \boldsymbol{\Sigma}_\pi\right)\right] \tag{59}$$

### 3.5 Posterior inference

### 3.5.1 Joint posterior

$$p(x) = \frac{m_{\mathbf{y}}^L(x)\pi(x)}{\mathbb{E}[Z|\mathbf{y}]} \tag{60}$$

### 3.5.2 Marginal posterior

The marginal posterior can be on obtained from Gaussian mixture form of joint posterior. Thanks to the Gaussianity, marginal posterior can be easily derived by extracting the d-th element of matrices in the following mixture of Gaussians.

$$p(x) = \frac{\alpha}{\mathbb{E}[Z|\mathbf{y}]} + \sum_{i,j} w_{ij}^p \mathcal{N}(x_*; \mu_p, \boldsymbol{\Sigma}_p) \tag{61}$$

where
$w_{ij}^p = \frac{w_{ij}^m}{\mathbb{E}[Z|\mathbf{y}]}\mathcal{N}\left(\frac{X_i + X_j}{2}; \mu_\pi, \frac{\mathbf{W}}{2} + \boldsymbol{\Sigma}_\pi\right)$
$\boldsymbol{\Sigma}_p = (2\mathbf{W}^{-1} + \boldsymbol{\Sigma}_\pi^{-1})^{-1}$
$\mu_p = \boldsymbol{\Sigma}_p(\mathbf{W}^{-1}(X_i + X_j) + \boldsymbol{\Sigma}_\pi^{-1}\mu_\pi)$

### 3.5.3 Conditional posterior

The conditional posterior $p(x; d = d \,|\, d = \mathbf{D} \setminus \mathbf{D}(\geq d))$ can be derived from the Gaussian mixture form of joint posterior. We can obtain the conditional posterior via applying the following relationship to each Gaussian: Assume $\mathbf{x} \sim \mathcal{N}(\mathbf{x}; \boldsymbol{\mu}, \boldsymbol{\Sigma})$ where

$$\mathbf{x} = \begin{bmatrix} \mathbf{x}_a \\ \mathbf{x}_b \end{bmatrix} \qquad \boldsymbol{\mu} = \begin{bmatrix} \boldsymbol{\mu}_a \\ \boldsymbol{\mu}_b \end{bmatrix} \qquad \boldsymbol{\Sigma} = \begin{bmatrix} \boldsymbol{\Sigma}_a & \boldsymbol{\Sigma}_c \\ \boldsymbol{\Sigma}_c^\top & \boldsymbol{\Sigma}_b \end{bmatrix} \tag{62}$$

Then

$$p(\mathbf{x}_a)|p(\mathbf{x}_b) = \mathcal{N}(\mathbf{x}_a; \hat{\boldsymbol{\mu}}_a, \hat{\boldsymbol{\Sigma}}_a) \qquad \begin{cases} \hat{\boldsymbol{\mu}}_a = \boldsymbol{\mu}_a + \boldsymbol{\Sigma}_c\boldsymbol{\Sigma}_b^{-1}(\mathbf{x}_b - \boldsymbol{\mu}_b) \\ \hat{\boldsymbol{\Sigma}}_a = \boldsymbol{\Sigma}_a - \boldsymbol{\Sigma}_c\boldsymbol{\Sigma}_b^{-1}\boldsymbol{\Sigma}_c^\top \end{cases} \tag{63}$$

$$p(\mathbf{x}_b)|p(\mathbf{x}_a) = \mathcal{N}(\mathbf{x}_b; \hat{\boldsymbol{\mu}}_b, \hat{\boldsymbol{\Sigma}}_b) \qquad \begin{cases} \hat{\boldsymbol{\mu}}_b = \boldsymbol{\mu}_b + \boldsymbol{\Sigma}_c\boldsymbol{\Sigma}_a^{-1}(\mathbf{x}_a - \boldsymbol{\mu}_a) \\ \hat{\boldsymbol{\Sigma}}_b = \boldsymbol{\Sigma}_b - \boldsymbol{\Sigma}_c\boldsymbol{\Sigma}_a^{-1}\boldsymbol{\Sigma}_c^\top \end{cases} \tag{64}$$

## 4 Uncertainty sampling

### 4.1 Analytical form of acquisiton function

### 4.1.1 Acquisiton function as Gaussian Mixture

We set the acquisition function $A(x)$ as the product of the variance and the prior. As is shown in Eq. (47), When we provide the predictive samples $x_*$:

$$A(x_*) = C_{\mathbf{y}}^{L}(x_*, x_*)\pi(x_*) \tag{65}$$

$$= C_{\mathbf{y}}^{L}(x_*)\pi(x_*) \tag{66}$$

$$= \left( \sum_{i,j} w_{ij}^{'v} \mathcal{N}\left(x_*; \frac{X_i + X_j}{2}, \frac{\mathbf{W}}{2}\right) - \sum_{i,j}\sum_{k,l} w_{ijkl}^{'vv} \mathcal{N}\left(x_*; \frac{X_i + X_j + X_k + X_l}{4}, \frac{\mathbf{W}}{4}\right) \right) \tag{67}$$

$$\cdot \mathcal{N}\left(x_*; \mu_\pi, \boldsymbol{\Sigma}_\pi\right) \tag{68}$$

$$= \sum_{i,j} w_{ij}^{'A} \mathcal{N}\left(x_*; \mu_{ij}^{A}, \boldsymbol{\Sigma}_A\right) - \sum_{i,j}\sum_{k,l} w_{ijkl}^{'AA} \mathcal{N}\left(x_*; \mu_{ijkl}^{AA}, \boldsymbol{\Sigma}_{AA}\right) \tag{69}$$

where
$\boldsymbol{\Sigma}_A = (2\mathbf{W}^{-1} + \boldsymbol{\Sigma}_\pi^{-1})^{-1}$
$\mu_{ij}^{A} = \boldsymbol{\Sigma}_A \left(\mathbf{W}^{-1}(X_i + X_j) + \boldsymbol{\Sigma}_\pi^{-1}\mu_\pi\right)$
$w_{ij}^{'A} = w_{ij}^{'v} \mathcal{N}\left(\frac{X_i + X_j}{2}; \mu_\pi, \frac{\mathbf{W}}{2} + \boldsymbol{\Sigma}_\pi\right)$

$\boldsymbol{\Sigma}_{AA} = (4W^{-1} + \boldsymbol{\Sigma}_\pi^{-1})^{-1}$
$\mu_{ijkl}^{AA} = \boldsymbol{\Sigma}_A \left(\mathbf{W}^{-1}(X_i + X_j + X_k + X_l) + \boldsymbol{\Sigma}_\pi^{-1}\mu_\pi\right)$
$w_{ijkl}^{'AA} = w_{ijkl}^{'vv} \mathcal{N}\left(\frac{X_i + X_j + X_k + X_l}{4}; \mu_\pi, \frac{\mathbf{W}}{4} + \boldsymbol{\Sigma}_\pi\right)$

### 4.1.2 Normalising constant and PDF

The normalising constant can be obtained via the integral:

$$\begin{aligned} Z_A &= \int A(x_*)dx_* \\ &= \sum_{i,j} w_{ij}^{'A} - \sum_{i,j}\sum_{k,l} w_{ijkl}^{'AA} \end{aligned} \tag{70}$$

Thus, the normalised acquisition function as PDF $p_A$ is as follows:

$$p_A(x_*) = \tilde{A}(x_*) \tag{71}$$

$$= \frac{C_{\mathbf{y}}^{L}(x_*)\pi(x_*)}{Z_A} \tag{72}$$

$$= \sum_{i,j} w_{ij}^{A} \mathcal{N}(x_*; \mu_{ij}^{A}, \boldsymbol{\Sigma}_A) - \sum_{i,j}\sum_{k,l} w_{ijkl}^{AA} \mathcal{N}(x_*; \mu_{ijkl}^{AA}, \boldsymbol{\Sigma}_{AA}) \tag{73}$$

where $w_{ij}^{A} = w_{ij}^{'A}/Z_A$
$w_{ijkl}^{AA} = w_{ijkl}^{'AA}/Z_A$

### 4.1.3 Factorisation trick

The factorisation trick is set in the conditions where the likelihood is $|\tilde{\ell}(x)|$, the distribution of interest $f(x)$ is $|\tilde{\ell}(x)|\pi(x)$, and the acquiition function is $\tilde{C}_{\mathbf{y}}(x, x)\pi(x)$. We will derive the Gaussian mixture form of this acquisition function.

$$A(x) = \tilde{C}_{\mathbf{y}}(x, x)\pi(x) \tag{74}$$

$$= \frac{v}{\sqrt{|2\pi\mathbf{W}|}} \mathcal{N}(x; \mu_\pi, \boldsymbol{\Sigma}_\pi) - v^2 \sum_{ij} \boldsymbol{\Omega}_{ij} \mathcal{N}(x; \mu^{f}, \boldsymbol{\Sigma}^{f}) \tag{75}$$

where
$\boldsymbol{\Sigma}^{f} = \left(2\mathbf{W}^{-1} + \boldsymbol{\Sigma}_\pi^{-1}\right)^{-1}$
$\boldsymbol{\mu}^{f} = \boldsymbol{\Sigma}^{f} \left(\mathbf{W}^{-1}(X_i + X_j) + \boldsymbol{\Sigma}_\pi^{-1}\mu_\pi\right)$

Then, normalising constant is:

$$Z_A^f = \int A(x)dx = \frac{v}{\sqrt{|2\pi\mathbf{W}|}} - v^2 \sum_{ij} \mathbf{\Omega}_{ij} \tag{76}$$

Therefore, the acquisition function as a probability distribution function $p_A(x)$ is:

$$p_A(x) = \frac{v}{Z_A^f\sqrt{|2\pi\mathbf{W}|}}\mathcal{N}(x; \mu_\pi, \mathbf{\Sigma}_\pi) - \frac{v^2}{Z_A^f} \sum_{ij} \mathbf{\Omega}_{ij}\mathcal{N}(x; \mu^f, \mathbf{\Sigma}^f) \tag{77}$$

## 4.2 Efficient sampler

### 4.2.1 Acquisition function as sparse Gaussian mixture sampler

Eq. (77) clearly explains the acquisition function can be written as a Gaussian mixture, but it also contains negative components. The first term is obviously positive, and the second term is a mixture of positive and negative components. The condition where the second term becomes positive is $\mathbf{\Omega}_{ij} < 0$. By checking the negativity of the element $\mathbf{\Omega}_{ij}$, we can reduce the number of components by half on average. Then, when we consider sampling from this non-negative acquisition function, the following steps will be performed: First, we sample the index of the component from weighted categorical distribution $\Pi(x)$, and the weights are the one in Eq. (77). Then, we sample from the normal distribution that has the same index identified in the first process. These sampling will be repeated until the accumulated number of the sample reaches the same as the recombination sample size $N$. This means the component whose weight is lower than $1/N$ is unlikely to be sampled even once. Therefore, we can dismiss these components with the threshold of $1/N$. Interestingly, the weights of Gaussians vary exponentially. The reduced number of Gaussians is much lower than $n^2$. As such, we can construct the efficient sparse Gaussian mixture sampler of the acquisition function $p_A'(x)$.

### 4.2.2 Sequential Monte Carlo

Recall from the Eqs (8) - (10) in the main paper, we wish to sample from $g(x) = (1-r)\pi(x)+rp_A(x)$. We have the efficient sampler $p_A'(x)$, but $p_A'(x) \neq p_A(x)$ because $p_A'(x)$ is the function which is constructed from only positive components of $p_A(x)$. Thus, we need to correct this difference via sequential Monte Carlo (SMC). The idea of SMC is simple:

1. sample $\mathbf{x} \sim p_A'(x)$, $\mathbf{x} \in \mathbb{R}^{rN}$
2. calculate weights $\mathbf{w}_{\mathrm{smc}} = p_A(\mathbf{x})/p_A'(\mathbf{x})$
3. resample from the categorical distribution of the index of $\mathbf{x}$ based on $\mathbf{w}_{\mathrm{smc}}$

If $p_A(\mathbf{x}) \approx p_A'(\mathbf{x})$, the rejected samples in the procedure 3 is minimised. As we formulate $p_A'(\mathbf{x})$ can approximate $p_A(\mathbf{x})$ well, the number of samples to be rejected is negligibly small. Thus, the number of samples from $p_A(x)$ is slightly smaller than $rN$. The number of samples for $\pi(x)$ in $g(x)$ is adjusted to this fluctuation to keep the partition ratio $r$.

## 5 Other BQ modelling

## 5.1 Non-Gaussian Prior

Non-Gaussian prior distributions can be applied via importance sampling.

$$\int \ell(x)\pi(x) = \int \ell(x)\frac{\pi(x)}{g(x)}g(x)dx \tag{78}$$

$$= \int \ell'(x)g(x)dx \tag{79}$$

where $\pi(x)$ is the arbitrary prior distribution of interest, $g(x)$ is the proposal distribution of Gaussian (mixture), $\ell'(x) = \ell(x)\pi(x)/g(x)$ is the modified likelihood. Then, we set the two independent GPs on each of $\ell(x)$ and $\ell'(x)$. Then, both the model evidence $Z = \int \ell'(x)g(x)dx$, and the posterior $p(x) = \ell(x)\pi(x)/Z$ becomes analytical.

## 5.2 Non-Gaussian kernel

WSABI-BQ methods are limited to the squared exponential kernel in the likelihood modelling. However, other BQ modelling permits the selection of different kernels. For instance, there are the existing works on tractable BQ modelling with kernels of Matérn [7], Wendland [16], Gegenbauer [7], Trigonometric (Integration by parts), splines [20] polynomial [6], and gradient-based kernel [15]. See details in [7].

## 5.3 RCHQ for Non-Gaussian prior and kernel

RCHQ permits the integral estimation via non-Gaussian prior and/or kernel without bespoke modelling like the above techniques.

$$\mathbf{X}_{\text{quad}}, \mathbf{w}_{\text{quad}} = \text{RCHQ}(\text{BQmodel}, \text{sampler}) \tag{80}$$

$$\mathbb{E}[\ell(x)\pi(x)] = \mathbf{w}_{\text{quad}} m_{\mathbf{y}}^L(\mathbf{X}_{\text{quad}}) \tag{81}$$

$$\mathbb{V}\text{ar}[\ell(x)\pi(x)] = \mathbf{w}_{\text{rec}}^\top C_{\mathbf{y}}^L(\mathbf{x}_{\text{rec}}, \mathbf{x}_{\text{rec}})\mathbf{w}_{\text{rec}} - 2\,\mathbf{w}_{\text{rec}}^\top C_{\mathbf{y}}^L(\mathbf{x}_{\text{rec}}, \mathbf{x}_{\text{quad}})\mathbf{w}_{\text{quad}} + \mathbf{w}_{\text{quad}}^\top C_{\mathbf{y}}^L(\mathbf{x}_{\text{quad}}, \mathbf{x}_{\text{quad}})\mathbf{w}_{\text{quad}} \tag{82}$$

## 5.4 Vanilla BQ model (VBQ)

### 5.4.1 Expectation

$$\int m_{\ell_0}(x)\pi(x)dx = v \int \mathcal{N}(x; \mathbf{X}, \mathbf{W})\mathcal{N}(x; \mu_\pi, \mathbf{\Sigma}_\pi)dx\boldsymbol{\omega} \tag{83}$$

$$= v\mathcal{N}(\mathbf{X}; \mu_\pi, \mathbf{W} + \mathbf{\Sigma}_\pi)\boldsymbol{\omega} \tag{84}$$

$$\tag{85}$$

### 5.4.2 Acquisition function

$$A_{\text{unnormalised}}(x) = C(x, x)\pi(x) \tag{86}$$

$$= K(x, x)\pi(x) - K(x, \mathbf{X})K(\mathbf{X}, \mathbf{X})^{-1}K(\mathbf{X}, x)\pi(x) \tag{87}$$

$$= \mathcal{N}(x; x, \mathbf{W})\mathcal{N}(x; \mu_\pi, \mathbf{\Sigma}_\pi) - v^2\mathcal{N}(x; \mu_\pi, \mathbf{\Sigma}_\pi)\mathcal{N}(x; \mathbf{X}, \mathbf{W})K(\mathbf{X}, \mathbf{X})^{-1}\mathcal{N}(x; \mathbf{X}, \mathbf{W})^\top \tag{88}$$

$$= \frac{v}{\sqrt{|2\pi\mathbf{W}|}}\mathcal{N}(x; \mu_\pi, \mathbf{\Sigma}_\pi) - v^2\sum_{i,j}\Omega_{ij}\mathcal{N}(\mu_\pi; X_i, \mathbf{W} + \mathbf{\Sigma}_\pi)\mathcal{N}(x; X_i', \mathbf{W}')\mathcal{N}(x; X_j, \mathbf{W}) \tag{89}$$

$$= \frac{v}{\sqrt{|2\pi\mathbf{W}|}}\mathcal{N}(x; \mu_\pi, \mathbf{\Sigma}_\pi) - v^2\sum_{i,j}\Omega_{ij}\mathcal{N}(\mu_\pi; X_i, \mathbf{W} + \mathbf{\Sigma}_\pi)\mathcal{N}(X_j; X_i', \mathbf{W} + \mathbf{W}')\mathcal{N}(x; X_{ij}'', \mathbf{W}'') \tag{90}$$

$$= \frac{v}{\sqrt{|2\pi\mathbf{W}|}}\mathcal{N}(x; \mu_\pi, \mathbf{\Sigma}_\pi) - \sum_{i,j}w_{ij}\mathcal{N}(x; X_{ij}'', \mathbf{W}'') \tag{91}$$

$$\tag{92}$$

where

$$\Omega_{ij} := K(\mathbf{X}, \mathbf{X})^{-1} \tag{93}$$

$$w_i := v^2\Omega_{ij}\mathcal{N}(\mu_\pi; X_i, \mathbf{W} + \mathbf{\Sigma}_\pi)\mathcal{N}(X_j; X_i', \mathbf{W} + \mathbf{W}') \tag{94}$$

$$\mathbf{W}' = (\mathbf{W}^{-1} + \mathbf{\Sigma}_\pi^{-1})^{-1} \tag{95}$$

$$X_i' = \mathbf{W}'(\mathbf{W}^{-1}X_i + \mathbf{\Sigma}_\pi^{-1}\mu_\pi) \tag{96}$$

$$\mathbf{W}'' = (\mathbf{W}'^{-1} + \mathbf{W}^{-1})^{-1} \tag{97}$$

$$X_{ij}'' = \mathbf{W}''(\mathbf{W}'^{-1}X_i' + \mathbf{W}^{-1}X_j) \tag{98}$$

$$\tag{99}$$

$$\tag{100}$$

Then, the normalised acquisition function $p_A(x)$ as a probability distribution is as follows:

$$P_A(x) := A_{\text{unnormalised}}(x)/Z_A \tag{101}$$

$$= \frac{v}{Z_A \sqrt{|2\pi\mathbf{W}|}} \mathcal{N}(x; \mu_\pi, \mathbf{\Sigma}_\pi) - \sum_{i,j} \frac{w_i}{Z_A} \mathcal{N}(x; X_{ij}'', \mathbf{W}'') \tag{102}$$

$$\tag{103}$$

where

$$Z_A = \int A_{\text{unnormalised}}(x)dx \tag{104}$$

$$= \frac{v}{\sqrt{|2\pi\mathbf{W}|}} \int \mathcal{N}(x; \mu_\pi, \mathbf{\Sigma}_\pi)dx - v^2 \sum_{i,j} \Omega_{ij} \mathcal{N}(\mu_\pi; X_i, \mathbf{W} + \mathbf{\Sigma}_\pi) \int \mathcal{N}(x; X_i', \mathbf{W}')\mathcal{N}(x; X_j, \mathbf{W})dx \tag{105}$$

$$= \frac{v}{\sqrt{|2\pi\mathbf{W}|}} - v^2 \sum_{i,j} \Omega_{ij} \mathcal{N}(\mu_\pi; X_i, \mathbf{W} + \mathbf{\Sigma}_\pi)\mathcal{N}(X_j; X_i', \mathbf{W} + \mathbf{W}') \tag{106}$$

$$\tag{107}$$

## 5.5 Log-GP BQ modelling (BBQ)

### 5.5.1 BBQ modelling

The doubly-Bayesian quadrature (BBQ) is modelled with log-warped GPs as follows (see details in the paper [17]):

**Set three GPs**

$$p(\ell_0|\mathbf{D}) \sim \mathcal{GP}(\ell_0; m_{\ell_0}(x), C_{\ell_0}(x, x')) \tag{108}$$
$$p(\log \ell_0|\mathbf{D}) \sim \mathcal{GP}(\log \ell_0; m_{\log \ell_0}(x), C_{\log \ell_0}(x, x')) \tag{109}$$
$$p(\Delta_{\log \ell_0}|\mathbf{D}) \sim \mathcal{GP}(\Delta_{\log \ell_0}; m_\Delta(x), C_\Delta(x, x')) \tag{110}$$
$$\tag{111}$$

**Definitions**

$$\exp(\log \ell(x)) \approx \exp(\log \ell_0(x)) + \exp(\log \ell_0(x))(\log \ell(x) - \log \ell_0(x)) \tag{112}$$
$$\ell_0 := m_{\ell_0} \tag{113}$$
$$\Delta_{\log \ell_0} := m_{\log \ell_0} - \log \ell_0 = m_{\log \ell_0} - \log(m_{\ell_0}) \tag{114}$$
$$m_\ell = m_{\ell_0} + m_{\ell_0} m_\Delta(x) \tag{115}$$

**Expectation**

$$\mathbb{E}[Z|\mathbf{D}] = \int m_{\ell_0}(x)\pi(x)dx + \int m_{\ell_0}(x)m_\Delta(x)\pi(x)dx \tag{116}$$

$$\tag{117}$$

The first term is as follows:

$$\int m_{\ell_0}(x)\pi(x)dx = v \int \mathcal{N}(x; \mathbf{X}, \mathbf{W})\mathcal{N}(x; \mu_\pi, \mathbf{\Sigma}_\pi)dx\boldsymbol{\omega} \tag{118}$$

$$= v\mathcal{N}(\mathbf{X}; \mu_\pi, \mathbf{W} + \mathbf{\Sigma}_\pi)\boldsymbol{\omega} \tag{119}$$

$$\tag{120}$$

where

$$\boldsymbol{\omega} = K(\mathbf{X}, \mathbf{X})^{-1}\ell_0(\mathbf{X}) \tag{121}$$
$$K(x, \mathbf{X}) = v\mathcal{N}(x; \mathbf{X}, \mathbf{W}) \tag{122}$$

The second term is as follows:

$$\int m_{\ell_0}(x)\Delta_{\log \ell_0}(x)p(x)dx \tag{123}$$

$$= vv^{\Delta}\boldsymbol{\omega}^{\top}\int \mathcal{N}(x;\mathbf{X},\mathbf{W})^{\top}\mathcal{N}(x;\mathbf{X}^{\Delta},\mathbf{W}^{\Delta})\mathcal{N}(x;\mu_{\pi},\boldsymbol{\Sigma}_{\pi})dx\boldsymbol{\omega}^{\Delta} \tag{124}$$

$$= vv^{\Delta}\boldsymbol{\omega}^{\top}\mathcal{N}(\mathbf{X}^{\top}-\mathbf{X}^{\Delta},\mathbf{0},\mathbf{W}+\mathbf{W}^{\Delta})\int \mathcal{N}(x;\boldsymbol{\mu}^{\Delta},\boldsymbol{\Sigma}^{\Delta})\mathcal{N}(x;\mu_{\pi},\boldsymbol{\Sigma}_{\pi})dx\boldsymbol{\omega}^{\Delta} \tag{125}$$

$$= vv^{\Delta}\boldsymbol{\omega}^{\top}\mathcal{N}(\mathbf{X}^{\top}-\mathbf{X}^{\Delta},\mathbf{0},\mathbf{W}+\mathbf{W}^{\Delta})\mathcal{N}(\boldsymbol{\mu}^{\Delta};\mu_{\pi},\boldsymbol{\Sigma}_{\pi}+\boldsymbol{\Sigma}^{\Delta})\boldsymbol{\omega}^{\Delta} \tag{126}$$

$$\tag{127}$$

where

$$\boldsymbol{\omega}^{\Delta} = K(\mathbf{X}^{\Delta},\mathbf{X}^{\Delta})^{-1}\Delta_{\log \ell_0}(\mathbf{X}^{\Delta}) \tag{128}$$

$$\boldsymbol{\mu}^{\Delta} = [\mathbf{W}^{-1}+\mathbf{W}^{\Delta,-1}]^{-1}(\mathbf{W}^{-1}\mathbf{X}+\mathbf{W}^{\Delta,-1}\mathbf{X}^{\Delta}) \tag{129}$$

$$\boldsymbol{\Sigma}^{\Delta} = [\mathbf{W}^{-1}+\mathbf{W}^{\Delta,-1}]^{-1} \tag{130}$$

$$\tag{131}$$

$\mathbf{X}^{\Delta}$ is the observed data for the correlation factor $\Delta_{\log \ell_0}$, which includes not only $\mathbf{X}$ but also the additional data points via $m_{\log \ell_0} - \log(m_{\ell_0})$, with GPs calculation.

### 5.5.2 Sampling for BBQ

We apply BASQ-VBQ sampling scheme for log-GP $\log \ell_0$, then calculate the others as post-process. Therefore, the sampling cost is similar to the VBQ, whereas the integral estimation as post-process is more expensive than VBQ.

## 6 Experimental details

### 6.1 Synthetic problems

#### 6.1.1 Quadrature hyperparameters

The initial quadrature hyperparameters are as follows:
A kernel length scale $l = 2$
A kernel variance $v' = 2$
Recombination sample size $N = 20,000$
Nyström sample size $M = N/100$
Supersample ratio $r_{\text{super}} = 100$
Proposal distribution $g(x)$ partition ratio $r = 0.5$

The supersample ratio $r_{\text{super}}$ is the ratio of supersamples for SMC sampling of acquisition function against the recombination sample size $N$.

A kernel length scale and a kernel variance are important for selecting the samples in the first batch. Nevertheless, these parameters are updated via type-II MLE optimisation after the second round. Nyström sample size must be larger than the batch size $n$, and the recombination sample size is preferred to satisfy $N \gg M$. Larger $N$ and $M$ give more accurate sample selection via kernel quadrature. However, larger subsamples result in a longer wall-time. We do not need to change the values as long as the integral converged to the designated criterion. When longer computational time is allowed, or likelihood is expensive enough to regard recombination time as negligible, larger $N$, $M$ will give us a faster convergence rate.

The partition ratio $r$ is the only hyperparamter that affects the convergence sensitively. The optimal value depends the integrand and it is challenging to know the optimal value before running. As we derived in Lemma 1, $\sqrt{C_{\mathbf{y}}^{L}}\pi(x)$ gives the optimal upper bound. $r = 0.5$ is a good approximation of this optimal proposal distribution: $g(x) = (1-r)\pi(x) + rC_{\mathbf{y}}^{L}\pi(x) = \left\{(1-r) + rC_{\mathbf{y}}^{L}\right\}\pi(x)$. Here,

the linearisation gives the approximation $\sqrt{C_{\mathbf{y}}^L} = \sqrt{1 + (C_{\mathbf{y}}^L - 1)} \approx 1 + \frac{C_{\mathbf{y}}^L - 1}{2} = 0.5 + 0.5 C_{\mathbf{y}}^L$. Therefore, $(0.5 + 0.5 C_{\mathbf{y}}^L)\pi(x) \approx \sqrt{C_{\mathbf{y}}^L}\pi(x)$. Thus, $r = 0.5$ is a safe choice.

### 6.1.2 Gaussian mixture

The likelihood function of the Gaussian mixture used in Figure 1 in the main paper is expressed as:

$$\ell_{\text{true}}(x) = \sum_{i=1}^{n} w_i \mathcal{N}(x; \mu_i, \boldsymbol{\Sigma}_i) \tag{132}$$

$$w_i = \mathcal{N}(\mu_i; \mu_\pi, \boldsymbol{\Sigma}_i + \boldsymbol{\Sigma}_\pi)^{-1} \tag{133}$$

$$Z_{\text{true}} = \int \ell_{\text{true}}(x)\pi(x)dx, \tag{134}$$

$$= \sum_{i=1}^{n} w_i \int \mathcal{N}(x; \mu_i, \boldsymbol{\Sigma}_i)\mathcal{N}(x; \mu_\pi, \boldsymbol{\Sigma}_\pi)dx \tag{135}$$

$$= \sum_{i=1}^{n} w_i \mathcal{N}(\mu_i; \mu_\pi, \boldsymbol{\Sigma}_i + \boldsymbol{\Sigma}_\pi) \tag{136}$$

$$= 1 \tag{137}$$

where
$\mu_\pi = \mathbf{0}$
$\boldsymbol{\Sigma}_\pi = 2\mathbf{I}$
$\pi(x) = \mathcal{N}(x; \mu_\pi, \boldsymbol{\Sigma}_\pi)$

The prior is the same throughout the synthetic problems.

### 6.1.3 Branin-Hoo function

The Branin-Hoo function in Figure 2 in the main paper is expressed as:

$$\ell_{\text{true}}(x) = \prod_{i=1}^{2} \frac{\left[\sin(x_i) + \frac{1}{2}\cos(3x_i)\right]^2}{(\frac{1}{2}x_i)^2 + \frac{3}{10}}, \quad x \in \mathbb{R}^2 \tag{138}$$

$$Z_{\text{true}} = \int \ell_{\text{true}}(x)\pi(x)dx \tag{139}$$

$$= 0.955728^2 \tag{140}$$

$$\approx 0.913416 \tag{141}$$

### 6.1.4 Ackley function

The Ackley function in Figure 2 in the main paper is expressed as:

$$\ell_{\text{true}}(x) = -20\exp\left(-\frac{1}{5}\sqrt{\frac{1}{2}\sum_{i=1}^{2}x_i^2}\right) + \exp\left(\frac{1}{2}\sum_{i=1}^{2}\cos(2\pi x_i)\right) + 20, \quad x \in \mathbb{R}^2 \tag{142}$$

$$Z_{\text{true}} = \int \ell_{\text{true}}(x)\pi(x)dx \tag{143}$$

$$\approx 5.43478 \tag{144}$$

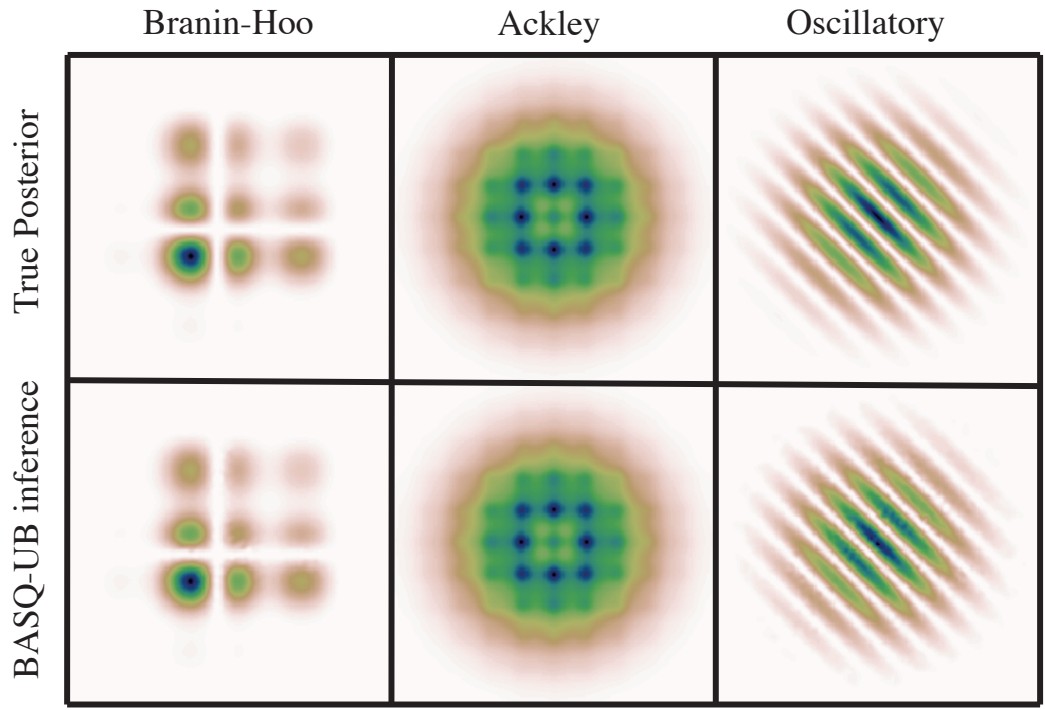

Figure 2: Performance comparison with N-dimensional Gaussian mixture likelihood function. (a) dimension study, (b) convergence rate, and (c) wall time vs MAE of integral. (a) varies from 2 to 16 dimensions, (b) and (c) are 10 dimensional Gaussian mixture.

### 6.1.5  Oscillatory function

The Oscillatory function in Figure 2 in the main paper is expressed as:

$$\ell_{\text{true}}(x) = \cos\left(2\pi + 5\sum_{i=1}^{2} x_i\right) + 1, \quad x \in \mathbb{R}^2 \tag{145}$$

$$Z_{\text{true}} = \int \ell_{\text{true}}(x)\pi(x)dx \tag{146}$$

$$= 1 \tag{147}$$

### 6.1.6  Additional experiments

**Dimensional study in Gaussian mixture likelihood**  Figure 2(a) shows the dimension study of Gaussian mixture likelihood. The BASQ and BQ are conditioned at the same time budget (200 seconds). The higher dimension gives a more inaccurate estimation. From this result, we recommend using BASQ with fewer than 16 dimensions.

**Ablation study**  We investigated the influence of the approximation we adopted using 10 dimensional Gaussian mixture likelihood. The compared models are as follows:

1. Exact sampler (without factorisation trick)
2. Provable recombination (without LP solver)

The exact sampler without the factorisation trick is the one that exactly follows the Eqs. (8) - (10) of the main paper. That is, the distribution of interest $f(x)$ is the prior $\pi(x)$. In addition, the kernel for the acquisition function is an unwarped $C_{\mathbf{y}}^L$, which is computationally expensive. Next, the provable recombination algorithm is the one introduced in [19] with the best known computational complexity. As explained in the Background section of the main paper, our BASQ implementation is based on an

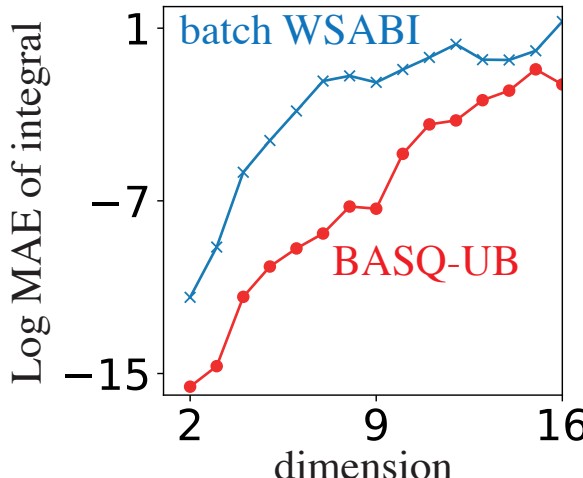

Figure 3: Qualitative evaluation of posterior inference in synthetic problems

LP solver (Gurobi [31] for this time) with empirically faster computational time. We compared the influence of these approximations.

Figure 2(b) illustrates that these approximations are not affecting the convergence rate in the sample efficiency. However, when compared to the wall-clock time (Figure 2(c)), the exact sampler without the factorisation trick is apparently slow to converge. Moreover, the provable recombination algorithm is slower than an LP solver implementation. Thus, the number of samples the provable recombination algorithm per wall time is much smaller than the LP solver. Therefore, our BASQ standard solver delivers solid empirical performance.

**Qualitative evaluation of posterior inference**    Figure 3 shows the qualitative evaluation of joint posterior inference after 200 seconds passed against the analytical true posterior. The estimated posterior shape is exactly the same as the ground truth.

## 6.2    Real-world problems

### 6.2.1    Battery simulator

**Background**    Single Particle Model with electrolyte dynamics (SPMe) is a commonly-used lithium-ion battery simulator to predict the voltage response at given excitation current time-series data. Estimating SPMe parameters from observations are well known for ill-conditioned problem because this model is overparameterised [5]. In the physical model, we need to separate the anode and cathode internal states to represent actual cell components. However, when it comes to predicting the voltage response, this separation into two components is redundant. Except for extreme conditions such as low temperature, most voltage responses can be expressed with a single component. Therefore, the parameters of cathode and anode often have a perfect negative correlation, meaning an arbitrary combination of cathode and anode parameters can reconstruct the exactly same voltage profile. As such, point estimation means nothing in these cases. Bayesian inference can capture this negative correlation as covariance. Therefore, Bayesian inference is a natural choice for parameter estimation in the battery simulator. Moreover, there are many plausible battery simulators with differing levels of approximation. Selecting the model satisfying both predictability and a minimal number of parameters is crucial for faster calculation, particularly in setting up the control simulator. Therefore, Bayesian model selection with model evidence is essential. The experimental setup is basically following [2].

**Problem setting**    We wish to infer the posterior distribution of 3 simulation parameters $(D_n, D_p, \sigma_n)$, where $D_n$ is the diffusivity on anode, $D_p$ is the diffusivity on cathode, $\sigma_n$ is the

noise variance of the observed data. We have the observed time-series voltage **y** and exciation profiles **i** as training dataset.

The parameter inference is modelled as follows:

$$y_* = \text{Sim}(x_*, i_*) \tag{148}$$

$$\pi(x_*) = \text{Lognormal}(x_*; \mu_\pi, \boldsymbol{\Sigma}_\pi) \tag{149}$$

$$\ell_{\text{true}}(x_*) = \mathcal{N}\left[\text{Sim}(x_*, \mathbf{i}); \mathbf{y}, \sigma_n \mathbf{I}\right] \tag{150}$$

where
$\mu_\pi = [1.38, 0, -20.25]$
$\boldsymbol{\Sigma}_\pi = \text{diag}([0.03, 0.001, 0.001])$
in the logarithmic space.

**Parameters**  The observed data **y** and **i** are generated by the simulator with multiharmonic sinusoidal excitation current defined as:

$$\mathbf{i} = 0.132671\left[\sin(1/5\pi t) + \sin(2\pi t) + \sin(20\pi t) + \sin(200\pi t)\right] \tag{151}$$

$$\mathbf{y} = \text{Sim}(x_{\text{true}}, \mathbf{i}) + \sqrt{\sigma_n}\mathcal{U}[0, 1] \tag{152}$$

where
$t$ is discretised for 10 seconds with the sampling rate of 0.00025 seconds, resulting in 40,000 data points.
$x_{\text{true}} = [\, \exp(1.361) \times 10^{-14}, \exp(0) \times 10^{-13}, \exp(-20.25) \times 10^{-10} \,]$

**Metric**  The posterior distribution is evaluated via RMSE between true and inferred conditional posterior on each parameter. The RMSE is calculated on 50 grid samples for each dimension so as to slice the maximum value of the joint posterior. Each 50 grid samples are equally-spaced and bounded with the following boundaries:
bounds $= [1.1, 1.7], [-0.075, 0.08], [-20.3, -20.2]$
where the boundaries are given by [lower, upper] in the logarithmic space.

### 6.2.2 Phase-field model

**Background**  The PFM is a flexible time-evolving interfacial physical model that can easily incorporate the multi-physical energy [13]. In this dataset, the PFM is applied to the simulation of spinodal decomposition, which is the self-organised nanostructure in the bistable Fe-Cr alloy at high temperatures. Spinodal decomposition is an inherently stochastic process, making characterisation challenging [14]. Therefore, Bayesian model selection is promising for estimating its parameter and determining the model physics component.

**Problem setting**  We wish to infer the posterior distribution of 4 simulation parameters $(T, L_{cT}, n_B, L_g)$, where $T$ is the temperature, $L_{cT}$ is the interaction parameter that defines the interaction between composition and temperature, $n_B$ is the number of Bohr magnetons per atom, and $L_g$ is the gradient energy coefficient. We have the observed time-series 2-dimensional images **y**.

The parameter inference is modelled as follows:

$$y_* = \text{Sim}(x_*) \tag{153}$$

$$\pi(x_*) = \text{Lognormal}(x_*; \mu_\pi, \boldsymbol{\Sigma}_\pi) \tag{154}$$

$$\ell_{\text{true}}(x_*) = \mathcal{N}\left[\text{Sim}(x_*); \mathbf{y}, \sigma_n \mathbf{I}\right] \tag{155}$$

where
$\sigma_n = 10^{-4}$
$\mu_\pi = [1.91, 0.718, 0.798, 0.693]$
$\boldsymbol{\Sigma}_\pi = \text{diag}([0.0003, 0.00006, 0.0001, 0.0001])$
in the logarithmic space.

**Parameters**  The observed data $\mathbf{y}$ is generated by the simulator defined as:

$$\mathbf{y} = \mathrm{Sim}(x_{\mathrm{true}}) + \sqrt{\sigma_n}\,\mathcal{U}[0,1] \tag{156}$$

where
$\mathbf{y}$ is discretised in both spatially and time-domain. Time domain is discretised for 5000 seconds with the sampling rate of 1 seconds, resulting in 5,000 data points. 2-dimensional space is discretised for $64 \times 64$ nm$^2$, with $64 \times 64$ nm$^2$ pixels. The total data points are $64 \times 64 \times 5,000 = 20,480,000$. $x_{\mathrm{true}} = [\ \exp(1.90657514) \times 10^2,\ \exp(0.71783979) \times 10^4,\ \exp(0.7975072),\ \exp(0.69314718) \times 10^{-15}\ ]$

**Metric**  The posterior distribution is evaluated via RMSE between true and inferred conditional posterior on each parameter. The RMSE is calculated on 50 grid samples for each dimension so as to slice the maximum value of the joint posterior. Each 50 grid samples are equally-spaced and bounded with the following boundaries:
bounds $= [1.87, 1.94], [0.69, 0.73], [0.77, 0.83], [0.68, 0.73]$
where the boundaries are given by [lower, upper] in the logarithmic space.

### 6.2.3  Hyperparameter marginalisation of hierarchical GP

**Background**  The hierarchical GP model was designed for analysing the large-scale battery time-series dataset from solar off-grid system field data all over the African continent [1]. The field data contains the information of time-series operating conditions $(I, T, V)$, where $I$ is the excitation current, $T$ is the temperature, and $V$ is the voltage. We wish to estimate the state of health (SoH) from these field data, achieving the preventive battery replacement before it fails for the convenience of those who rely on the power system for their living. However, estimating the state of health is challenging because the raw data $(I, T, V)$ is not correlated to the battery health. There are several definitions of SoH, but the internal resistance of a battery $R$ is adopted in [1]. In the usual circuit element, resistance can be easily calculated from $R = V/I$ via Ohm's law. However, the battery internal resistance $R$ is way more complex. Battery internal resistance $R$ is a function of $(t, I, T, c)$, where $t$ is time, $c$ is the acid concentration. Furthermore, there are two factors of resistance variation; ionic polarisation and aging. To incorporate these physical insights to the machine learning model, [1] is adopted the hierarchical GP model. First, they adopted the additive kernel of a squared exponential kernel and a Wiener velocity kernel to divide the ionic polarisation effect and aging effect. Second, they adopted the hierarchical GPs to model $V$ to divide into $R$-dependent GP and non-$R$-dependent GP to incorporate the Open Circuit Voltage-State of Charge (OCV-SOC) relationship.

**Problem setting**  We wish to infer the hyperposterior distribution of 5 GP hyperparameters $(l_T, l_I, l_c, \sigma_0, \sigma_1)$, where $l_T, l_I, l_c$ are the a squared exponential kernel lengthscale of temperature $T$, current $I$, and acid concentration $c$, respectively, and $\sigma_0, \sigma_1$ are the kernel variances of a squared exponential kernel and a Wiener velocity kernel, respectively. We have the observed time-series dataset of $(I, T, V)$ as $\mathbf{y}$.

The hyperposterior inference is based on the energy function $\Phi(x)$ (The details can be found in [1], Equation (15) in the Appendix information).

$$\Phi x = -\log p(\mathbf{y}|x) - \log p(x) \tag{157}$$

$$= -\log p(x) + \frac{1}{2}\sum_t \log |S_t(x)| + \frac{1}{2}\sum_t e_t^{\mathrm{T}} S_t(x)^{-1} e_t + \sum_t \frac{n_t}{2}\log 2\pi \tag{158}$$

where
$p(x) = \mathrm{Lognormal}(x_*; \mu_\pi, \mathbf{\Sigma}_\pi)$ is a hyperprior.
$e_t$ is the error vector for each charging segment.
$n_t$ is the number of observations in the charging segment.
$S_t(x)$ is the innovation covariance for the segment.
$\mu_\pi = [3.96, 1.94, 2.79, 2.26, 0.34]$
$\mathbf{\Sigma}_\pi = \mathrm{diag}([1, 1, 1, 1, 1])$
in the logarithmic space.

**Metric** The posterior distribution is evaluated via RMSE between true and inferred conditional posterior on each parameter. The RMSE is calculated on 50 grid samples for each dimension so as to slice the maximum value of the joint posterior. Each 50 grid samples are equally-spaced and bounded with the following boundaries:

bounds $= [-10, 10], [-10, 10], [-10, 10], [-10, 10], [-10, 10]$

where the boundaries are given by [lower, upper] in the logarithmic space.

# 7 Technical details; Q & A

**Q1: How does BASQ enable RCHQ to perform the batch selection?** A1: The trick that achieves parallelisation is the alternately subsampling in section 4.1, not RCHQ itself. While BQ aims to calculate the target integral $Z = \int \ell_{\text{true}}(x)\pi(x)dx$, RCHQ over a single iteration aims to calculate the empirical integral $Z = \int \ell(x)\pi_{\text{emp}}(x)dx$ over empirical measure defined by $N$ subsamples $\mathbf{X}_{\text{rec}}$. At each iteration, we greedily select the batch candidates via RCHQ that can minimise the integral variance over the current empirical measure. As we gather more observation data points and update the kernel (GP), the above two integrals approach the same. In other words, any KQ method, including kernel herding, can be applied to the batch selection via this alternately subsampling scheme. Secondly, such a dual quadrature scheme tends to be computationally demanding, but tractable computation and superb sample efficiency of RCHQ permit scalable batch parallelisation.

**Q2: Why does RCHQ outperform the kernel herding?** A2: The reason why RCHQ converges faster than herding is that RCHQ exploits more information than herding. While herding greedily optimises sequentially, RCHQ explicitly exploits the information of the spectral decay of the kernel and the probability distribution, both of which herding neglects. Exploiting the spectral decay corresponds to capturing the approximately finite dimensionality of the kernel. RCHQ adopts the Nyström method for its approximation. This convergence rate superiority can be confirmed in figure 2(a) in [11]. ("N. + emp + opt" refers to RCHQ.) While RCHQ exponentially decays, herding does not show such fast decay in the Gaussian setting. Therefore, BQ with RCHQ can converge faster than BQ with kernel herding, allowing scalable and sample-efficient batch selection.

**Q3: Are there some potential areas, if any, where the proposed method performs worse than existing ones?** A3: Probably yes, there is. The advantage of herding over RCHQ is the computation cost. In the small batch size setting, the difference in the level of convergence between herding and RCHQ is much smaller than in the large batch size $n$. Therefore, herding might perform better than RCHQ in the small batch with a very cheap likelihood case as herding might earn more samples than RCHQ. The comparison against other KQ methods is summarised in table 1 in [11]. RCHQ gives a small theoretical bound of the worst-case error with tractable computation cost compared to herding, DPP, CVS, and vanilla BQ.

**Q4: What are the pros and cons of RCHQ over the Determinantal Point Process (DPP)?** DPP considers the correlation correctly, whereas RCHQ assumes i.i.d. However, DPP requires prohibitive computation. Table 1 in [11] compares DPP-based KQ [4] and RCHQ ("N. + empirical" refers to RCHQ), which clearly shows that RCHQ provides not only tractable computation but also competitive theoretical bound of worst-case error with mathematical proof.

**Q5: Why Nyström? Other low-rank approximation possibilities?** A5: Because Nyström is advantageous to derive convergence based on spectral decay asymptotically and theoretically. The only requirement for the RCHQ is a finite sequence of good test functions, so finite dimensional approximations such as random Fourier features can also be used.

**Q6: Theorem 1 does not apply to WSABI-transformed BASQ but to a variant which uses vanilla BQ. Is that correct?** A6: Yes. Theorem 1 is under the assumptions which the BQ is modelled with vanilla BQ, without batch and hyperparameter updates. However, if we accept the linearisation of WSABI-L and assume that the $\ell(x)$ is (approximately) in the GP over the current iteration, the theoretical analysis is correct.

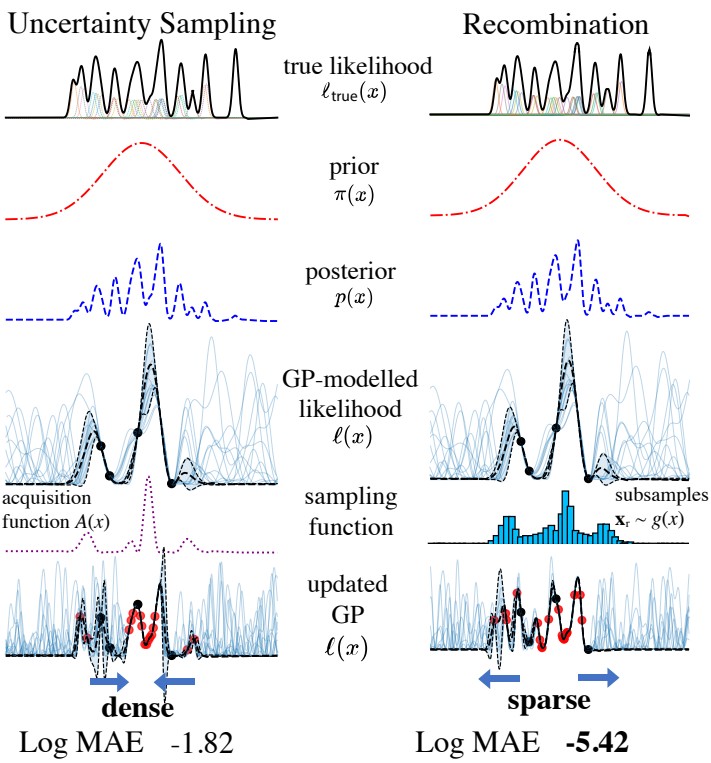

Figure 4: Performance comparison with 1-dimensional Gaussian mixture likelihood function.

**Q7: Why is the coefficient 0.8 used in the definition of alpha?**  A7: We inherit the coefficient of 0.8 from the original paper [10]. However, they said the performance is insensitive to the choice of this coefficient, so it is not limited to 0.8 in general.

**Q8: Can we apply WSABI for negative integral?**  A8: Yes. $\alpha$ can take negative value, so WSABI transformation can be applied to negative integral case when we apply WSABI to general function integration.

### 7.1  Detailed description of the difference between RCHQ and batch WSABI

Figure 4 shows the performance comparison between the kernel recombination (RCHQ) and the uncertainty sampling (batch WSABI). Firstly, the true likelihood $\ell_{\text{true}}$ was modelled with the mixture of one-dimensional Gaussians, which was generated under the procedure described in footnote 4 in the main paper. Prior $\pi(x)$ is a broader one-dimensional Gaussian, and the posterior $p(x)$ is also the mixture of Gaussians, thanks to the Gaussianity. While we can access the information of the prior distribution function $\pi(x)$, we cannot access the ones of the posterior $p(x)$ or true likelihood function $\ell_{\text{true}}$. However, we can query the true likelihood value at a given location $\ell_{\text{true}}(\mathbf{X}_{\text{quad}})$ with a large batch ($n = 16$ in this case). Now, we have four observations $n = 4$ with black dots. We have constructed the WSABI-L GPs with the given four observations $(\mathbf{X}, \mathbf{y})$. The mean dotted line represents the mean of posterior predictive distributions $m_{\mathbf{y}}^{L}(x)$, the blue shaded area shows the mean $\pm$ corrected variance $C_{\mathbf{y}}^{L}(x, x)\pi(x)$. A myriad of blue lines represents the functions sampled from GP $\ell \sim \mathcal{GP}(\ell; m_{\mathbf{y}}^{L}(x), C_{\mathbf{y}}^{L}(x, x))$. The above problem setting is shared with both algorithms.

On the one hand, batch WSABI adopts local penalisation with multi-start optimisation. The acquisition function $A(x)$ for batch WSABI is the uncertainty sampling $\mathbb{V}\text{ar}[\ell(x)\pi(x)] = \pi(x)^2 C_{\mathbf{y}}(x)m_{\mathbf{y}}(x)^2]$ as shown in purple dotted line. We can see four peaks corresponding to the positions of larger variance in WSABI-L GP. Multi-start optimisation generates 100 random samples from prior as multi-starting points, then run a gradient-based optimisation algorithm (L-BFGS) to find the maxima. Then, we take the largest point amongst the solutions. After taking the largest point,

we locally penalise the point with Lipschitz Cone. Then the largest peak is split into two peaks with smaller heights. Then, the multi-start optimisation will be applied again to find the next maxima. We will iterate this greedy selection for $n = 16$. The selected points are depicted with red dots. The WSABI-L GPs are updated with the given 16 observations. As we can see, the selected 16 batch candidates are concentrated around the largest peak in the acquisition function, resulting in a higher integration error. This is mainly because the local penalisation tends to aggregate around the large peak, where the newly generated penalised peaks still have significant heights. When compared the acquisition function $A(x)$ with the true posterior $p(x)$, we can find that the peak positions between $A(x)$ and $p(x)$ are not so correlated. In other words, local penalisation trusts the acquisition function too much, although the early-stage acquisition function $A(x)$ is not such a reliable information source. Moreover, the multi-start optimisation requires the optimisation loop per the number of seeds, which is computationally demanding. In addition, the possibility of finding the global optima becomes exponentially lower when the dimension is scaled up. Thus, the multi-start optimisation requires exponentially increasing the number of random seeds, although there are still no guarantees to find the global maxima of acquisition function $A(x)$. Therefore, batch WSABI is slow and inefficient in selecting the batch candidates.

On the other hand, RCHQ using the linearised IVR proposal distribution $g(x) = (1-r)\pi(x) + rA(x)$. This is mixed with the prior and GP variance, so it is less dependent on the early-stage $A(x)$. The subsampled histogram depicted with blue bars has similar peaks with $A(x)$, but still, there is room for the possibility of selecting other regions. Then, RCHQ constructs the empirical measure based on these $N$ subsamples and resamples $M$ samples for the Nyström method to construct the finite test functions. The test functions are applied to construct the metric to evaluate the worst-case error. Then, RCHQ selects $n$ batch candidates to minimise the worst-case error over the empirical measure with the kernel recombination. As we can see, the selected 16 batch candidates are sparser and well-captured the true likelihood peaks than local penalisation, resulting in a smaller integration error. Such subsampling is done faster than multi-start optimisation thanks to the efficient sampler, and recombination is also tractable with single LP solver iteration. As such, the RCHQ can select sparser candidates than local penalisation within more tractable computation time.