# OpenReview forum: "Fast Bayesian Inference with Batch Bayesian Quadrature via Kernel Recombination"
_NeurIPS.cc/2022/Conference — NeurIPS 2022 Accept_

### Official Review · Reviewer_CnRy · 2022-06-23

**Rating:** 5
**Confidence:** 4
**Soundness:** 3 good
**Presentation:** 3 good
**Contribution:** 2 fair

**Summary:**

This paper combines the batch-version of WSABI Bayesian quadrature method [30], which uses a square-root transform to encode non-negativity of the integrand function (in this case a likelihood function), with the kernel recombination method for sampling of quadrature points from [37] and the Nyström approximation of the kernel.

**Questions:**

ON THEORY:

1. Theorem 1, if correct (see the next two points), guarantees exponential convergence over a single iteration in the sense that the BQ integral variance decays exponentially fast. This is a consequence of the use of the extremely strong squared exponential GP prior. But no discussion or elaboration on the practical meaning of this statement can be found in the paper. Does the integral estimate produced by the method converge exponentially to the integral of the true likelihood function $\ell_\text{true}$? If so, what assumptions need to be imposed on $\ell_\text{true}$? To include an an unconditional statement that the proposed method "possesses a provably-exponential convergence rate" in the abstract is quite misleading.
2. It is claimed after Theorem 1 that the convergence of the variance is exponential if the kernel has exponentially decaying Mercer eigenvalues. But what about the second term on the right-hand side of Equation (12)? I do not see how this term is to decay exponentially.
3. Equation (12) in the supplement uses the property in Equation (6) of the main paper which says that BQ variance equals the minimal squared worst-case error in the RKHS of $K$. But this characterization only applies to the variance of the vanilla BQ, not the variance of WSABI. Therefore it seems to me that Theorem 1 does not apply to BASQ as proposed in Section 4 but to a variant which uses vanilla BQ instead of WSABI. Is that correct?

MINOR COMMENTS:

- p2, lines 38-39: "Choosing an appropriate kernel and prior makes the integrals analytical." I suppose this refers to the kernel mean embeddings etc. that need to be computed in BQ. However, how the sentence in inserted in the introduction makes it sound like the likelihood integrals that are being computed with BQ are available analytically. There is a similar issue on p2 line 54, "In particular, the integral becomes analytic...".
- Section 2: In this section, BQ is presented as a method specifically for computation of quantities related to posteriors ("The objective of BQ is to compute the following integrals and obtain surrogate function of posterior:") while in reality the integrals computed by BQ do not have to have anything to do with posteriors or Bayesian inference. It would be good to clarify this, e.g. saying that this is just what BQ is used for in this particular paper.
- p2, line 45: "following integrals" There is only one integral in Equation (1).
- p2, lines 60-61: "WSABI ... sets ..." It is unclear what this sentence is trying to say. What is being "set" to something?
- Section 2, in particular "Vanilla Bayesian quadrature": One unacquianted with GPs would probably have a very hard time trying to understand what happens in this section. For example, it is not said that the likelihood function is being modelled as GP(0, K) and that Equation (2) is the resulting posterior GP. The section also contains a lot of undefined notation.
- p2, line 62: Why is the coefficient 0.8 used in the definition of alpha?
- Section 2: I do not see why different notation for point sets is used in "Vanilla Bayesian quadrature" and "Kernel quadrature in general".
- p5, lines 160-1: A word must be missing ("selects sparse samples than batch WSABI"). Or "sparser"?
- Eq (9): dx in the denominator looks different than elsewhere in the paper.
- p6, line 178: Isn't $C_y^L$ the warped kernel?
- Lines 243-4: "that can earn per wall time" (?)
- Lines 246-7: "Still, the time becomes much longer when it increases for accuracy." (?)
- Line 248: "Yet, the original data size is roughly 1,000 times." (?)
- Line 268: "explorative UB proposal distribution exceeds" (?)
- Line 299: "It also works well with a moderate M in practise." What works?
- Ref [9]: "F. X. Briol" should be "F.-X. Briol"
- Ref [58]: "Sarkka" should be "Särkkä"
- Supplement, line 15 has \propto while Line 21 has \approx.


**Limitations:**

One of the main limitations of the proposed method (which is shared by all methods that utilize WSABI-BQ) is that it really only applies to $\pi$ being a mixture of Gaussian densities and $K$ the squared exponential kernel because in other cases the integrals of the functions in Equation (4) have not been computed analytically. This limitation should be stated more openly as at the moment the only allusion to it in the abstract and introduction is the sentence "Choosing an appropriate kernel and prior makes the integrals analytical." on p2, lines 38-9.

**Strengths And Weaknesses:**

I do not find this paper particularly exciting: the proposed BASQ method is a conceptually rather straightforward combination of a few existing methods. As far as I can see, the paper contains no interesting new ideas of any generality. The main contribution is that the proposed method outperforms some existing approaches in a number of examples. While the examples do seem impressive, I am not sure if this is sufficient and find the paper borderline, though tend to recommend rejection. However, I have no serious objections to acceptance if the consensus is that a contribution like this is sufficient for this conference.

I have certain concerns with the convergence theory in Section 6 and how it is advertised, which I detail in the next section of this review. Whether or not these concerns are valid has little impact on my assessment of this paper because I do not expect a paper like this to have any useful convergence theory.

The paper is generally reasonably well written and effortless to read. However, there are many unclear statements and undefined notation, examples of which I give in the next section. In general, I find that it would be helpful to replace many of the verbal descriptions with mathematics.

---

> ### Author Response · Authors · 2022-08-02
> **Authors' response summary: improved technical clarifications**
>
> Thank you for such a thorough review! We have updated the manuscript to address all points raised **with blue-coloured text**. In particular, we have amended the mathematically inaccurate phrases in the manuscript. Please see the details below:
>
> C1: An unconditional statement that the proposed method "possesses a provably-exponential convergence rate" in the abstract is quite misleading.
>
> A1: **Yes, we agree it is misleading to say the phrase "possesses a provably-exponential convergence rate.”** as if it sounds like there is proof that the whole algorithm converges exponentially. Therefore, we have deleted the phrase from the abstract. However, at the same time, we think this convergence analysis over a single iteration is the first step to being able to extend across steps as described in the section 7 discussion. Also, we believe it is an advancement for the NeurIPS community that we no longer assume that the space is compact, unlike the previous analysis [42] accepted by NeurIPS, as the Gaussian prior distribution is not compact obviously.
>
> C2: The second term on the right-hand side of Equation (12) does not seem to decay exponentially.
>
> A2: **Thank you. You are correct.** It was inaccurate to describe it as theoretically exponential, so we have deleted the phrase from the manuscript. Nevertheless, in some settings, the costly function evaluation allows us to regard the error term as negligible due to the computational resources at hand ($N$) for $n$. Furthermore, in figure 2(a) in the reference [37] ("N. + emp + opt" refers to RCHQ), empirical exponential decay was observed by optimising the weights (equivalent to Bayesian quadrature) after recombination for the empirical measure.
>
> C3: Theorem 1 does not apply to WSABI-transformed BASQ but to a variant which uses vanilla BQ. Is that correct?
>
> A3: **Strictly speaking, yes, it is** because the $\ell(x)$ itself is not in the GP. However, if we accept the linearisation of WSABI-L and assume that the $\ell(x)$ is (approximately) in the GP over the current iteration, the theoretical analysis is correct. The heuristic algorithm is built on this theoretical perspective.
>
> C4: One of the main limitations of the proposed method (which is shared by all methods that utilize WSABI-BQ) is that it really only applies to π being a mixture of Gaussian densities and K the squared exponential kernel because in other cases the integrals of the functions in Equation (4) have not been computed analytically.
>
> A4: **Partially correct.** It is correct that WSABI-BQ methods are limited to the squared exponential kernel in the likelihood modelling. However, we can take non-Gaussian prior distributions via importance sampling as described in lines 58-59. $\int \ell(x) \pi(x) = \int \ell(x) \pi(x) / g(x) g(x) dx = \int \ell^\prime(x) g(x) dx$, where $\pi(x)$ is the arbitrary prior distribution of interest, $g(x)$ is the proposal distribution of Gaussian (Mixture), $\ell^\prime(x) = \ell(x) \pi(x) / g(x)$ is the modified likelihood. The integral becomes analytical. We set the two independent GPs on each of $\ell(x)$ and $\ell^\prime(x)$. Then, both the model evidence $Z = \int \ell^\prime(x) g(x) dx$, and the posterior $p(x) = \ell(x) \pi(x) / Z$ becomes analytical. Furthermore, other BQ modelling permits the selection of different kernels. For instance, there are existing works on tractable BQ modelling with kernels of Matérn [S7], Wendland [S16], Gegenbauer [S7], Trigonometric (Integration by parts), splines [S19], polynomial [S6], and gradient-based kernel [S15].
> We believe the inherent originality of BASQ roots in the alternately subsampling between BQ and RCHQ, in which BQ modelling with the above kernels can be a variant of BASQ in future work.
>
> Note: We appreciate your minor comments with careful reading, all of which are very helpful and constructive. We have amended the corresponding parts in the manuscript.

---

> > ### Comment · Reviewer_CnRy · 2022-08-09
> > **Reviewer response**
> >
> > As the consensus appears to be that the technical improvements presented in the paper are sufficient for publication, I would be happy to increase my evaluation to "5. Borderline accept". However, I am unable to do this because the authors' response A3 indicates that Theorem 1 is flawed.
> >
> > A3: "Strictly speaking, yes, it is because the $\ell(x)$ itself is not in the GP. However, if we accept the linearisation of WSABI-L and assume that the $\ell(x)$ is (approximately) in the GP over the current iteration, the theoretical analysis is correct. The heuristic algorithm is built on this theoretical perspective."
> >
> > In that case Theorem 1 is either a) not a theorem and should not be presented as such or b) should contain a rigorous version of this assumption (I do not know what it should rigorously mean for "$\ell(x)$ to be (approximately) in the GP").
> >
> > ---
> >
> > Sidenote on A1: "Also, we believe it is an advancement for the NeurIPS community that we no longer assume that the space is compact, unlike the previous analysis [42] accepted by NeurIPS, as the Gaussian prior distribution is not compact obviously."
> >
> > Two sets of results may be of interest to you:
> >
> > The rate $N^{-1/2}$ for the expected integral variance on non-compact sets, as well as the proof given this paper, is not exactly novel (though perhaps this result has not appeared explicitly in ML literature). See Corollary 2.8 in https://core.ac.uk/outputs/153229370 (more standard and older sources ought to exist) where the assumption that $X$ is compact is not necessary for this particular result.
> >
> > In the case of Gaussian integration distribution and the Gaussian kernel exponential rates of convergence of the BQ integral variance or its expectation can be found (or follow immediately in combination with your Eq. (6)) in the following papers:
> >
> > https://proceedings.neurips.cc/paper/2019/hash/7012ef0335aa2adbab58bd6d0702ba41-Abstract.html [Theorem 1]
> >
> > https://link.springer.com/article/10.1007/s10543-011-0358-9 [Theorem 4.1]
> >
> > https://www.ams.org/journals/mcom/2017-86-304/S0025-5718-2016-03144-0/ [Theorem 3.2]
> >
> > https://www.ams.org/journals/mcom/2021-90-331/S0025-5718-2021-03659-5/ [Theorems 2.5 and 2.10]

---

> > > ### Author Response · Authors · 2022-08-09
> > > **Quick response to the point A3**
> > >
> > > Thank you for your reply and sorry fo the confusion. To clarify, we give a quick response to the point A3. In the assumption of Theorem, $\ell$ is explicitly stated as a function following a Gaussian process (probably using the same character $\ell$ caused the confusion). So the theorem itself is correct. However, in the algorithm, as we are warping the GP by using WSABI, the actual $\ell$ we want to estimate is modelled as a quadratic function of the function actually in a GP, so it doesn't generally follow a GP.
> > >
> > > To make it algorithmically tractable, in WSABI-L, we approximate the distribution of $\ell$ by some liniarisation to get another GP, which heuristically captures the behavior of $\ell$. Then we do our kernel recombination for this second GP.
> > >
> > > Our Theorem is true for a function actually in this second GP (though the statement is generally true for any GP), but in reality $\ell$ in our algorithm is not following this second GP, so it doesn't meet the assumption of the theorem. But "within the approximation error of WSABI-L heuristics", our theoretical analysis applies to the performance.
> > > We hope this clarifies the situation.

---

> > > ### Author Response · Authors · 2022-08-09
> > > **The manuscript updated**
> > >
> > > Thank you for your feedback! We have updated the manuscript to address the two points raised.
> > >
> > > C5: In that case Theorem 1 is either a) not a theorem and should not be presented as such or b) should contain a rigorous version of this assumption (I do not know what it should rigorously mean for "$\ell(x)$ to be (approximately) in the GP").
> > >
> > > A5: As we have already quickly answered, we added the following footnote 10 to notify this point;
> > > Here, although this theorem is generally true for any GP,  the GP given by WSABI-L $\ell_\text{WSABI-L}$ is a tractable approximation of the actual distribution of $\ell$. So $\ell$ we want to estimate in the algorithm is not following this (linearised) GP, but for a random function taken from the linearised GP, our theoretical analysis applies.
> > > Although we believe this statement is true, but we will happily move the whole section to supplementary and add the detailed explanation in the camera-ready version, if you believe this is misleading for readers and improve our paper quality. Due to the time zone difference, we are not sure if we can be awake until your response, but we promise that we will follow your thought in the camera-ready version.
> > >
> > > C6: Sidenote on A1
> > >
> > > A6: Thank you for introducing the existing works on mathematic field! We are ashamed that we have not noticed and not cited these works. We added the footnote 11 to introduce these papers. We will carefully read and learn from these works by the time when the camera-ready version.

---

> > > > ### Comment · Reviewer_CnRy · 2022-08-09
> > > > **Reviewer response**
> > > >
> > > > Thank you for some clarifications. So, if I understand correctly, this in the end boils down to the use of quite imprecise notation.
> > > >
> > > > In my opinion it is enough to explicitly mention somewhere near Theorem 1 that the variance on the left-hand side of Equation (12) is not the WSABI-L variance computed via Equations (4) and (3) as indicated on line 15 of Algorithm but the "vanilla BQ variance" that one computes using Equations (2) and (3). [I have hard time understanding what Footnote 10 tries to say. E.g., "$\ell$ we want to estimate in the algorithm". Is it not the model evidence defined by $\ell_\textup{true}$ that is being estimates in Algorithm 1?] Perhaps it might make sense to move or repeat in some form the sentence on lines 309-10 in Section 6 so that most of the shortcomings of the theory would be collected in one place.
> > > >
> > > > Given this, I feel that the current title of Section 6 is a bit misleading (and has likely contributed to some of my misunderstandings). E.g., just drop "for BASQ" from the title and in the first paragraph say that result is for e.g. a "simplified" version of BASQ.

---

> > > > > ### Author Response · Authors · 2022-08-09
> > > > > **Manuscript updated**
> > > > >
> > > > > Thank you for being constructive and supportive to improve our paper. Yes, you are correct on all points. We have updated the title and added the explanation on our assumption in the beginning of section 6 in lines 268-269. I find this explanation much easier to follow. Thank you! I also deleted the footnote 10 as it is confusing.

---

> > > > > > ### Comment · Reviewer_CnRy · 2022-08-09
> > > > > > **Reviewer response**
> > > > > >
> > > > > > I have updated my score to 5.

---

> > > > > > > ### Author Response · Authors · 2022-08-09
> > > > > > > **Acknowledgement**
> > > > > > >
> > > > > > > Thank you for increasing your score. We appreciate your help to improve readability. We believe our manuscript becomes much better than before.

---

### Official Review · Reviewer_Aavt · 2022-07-07

**Rating:** 6
**Confidence:** 3
**Soundness:** 2 fair
**Presentation:** 3 good
**Contribution:** 2 fair

**Summary:**

This paper proposes a novel quadrature procedure that combines elements of warped Bayesian quadrature and random convex hull quadrature. The resulting method enjoys benefits from both component aspects: the computational tractability of RCHQ and the integrand-modeling/active learning techniques of square-root warped BQ (WSABI). The authors empirically evaluate their method against a batch version of WSABI and various implements of nested sampling; they also provide some theoretical analysis on the convergence rate of their method.

**Questions:**

- Did the authors consider alternative low-rank approximation methods for the kernel matrix? Or is there something fundamental about the Nyström approximation that would invalidate this method if an alternative were used?

- Similarly, is there something about the WSABI-L approximation that makes it particularly well suited for the method proposed by the authors? Did they also experiment with the WSABI-M approximation or other previously-proposed transforms?

- In the empirical evaluation, are confidence intervals omitted for the real-word experiments or are they just imperceptibly small? In general, are there any meaningful takeaways from the fact that batch WSABI performs identically to the BASQ variants on the real-world benchmarks in terms of log MAE?

**Limitations:**

The authors' discussion of limitations in Section 7 is sufficiently detailed in my opinion.

**Strengths And Weaknesses:**

Overall, I think this paper presents some interesting ideas and represents a promising line of inquiry. However, in its current state, I believe there are some issues (detailed below) which keep this version from being a strong candidate for acceptance.

1. This work makes a litany of smaller, practical contributions which are certainly novel, although the core approach is a combination of existing ideas. However, I do think the methods described in Section 4.2 are sufficient in terms of getting this work above the bar in terms of originality.

2. My primary concern with this work is the level of analysis performed: given that this work combines many complex components, hyperparameters, design choices, heuristics, etc... an ablation study/sensitivity analysis of some sort seems warranted to assess which aspects are providing what benefit or if the sum is greater than the whole of its parts. Some notable components that I feel warrant a deeper dive include the hyperparameters M, N and r; the GP model on the integrand; sampling methodology (some of these are expanded on in my questions below).

Relatedly the subsample proposal distributions are somewhat poorly motivated heuristics: it is not clear to me why the uncertainty sampling acquisition function need be mixed with the prior to ensure global optimization. Calling this mixing the "integral variance reduction" distribution is also a bit strange: there is a well-known connection between DPPs and kernel quadrature and the correctly defined DPP would in some sense be the actual integral variance reducing distribution (although I suspect that sampling from this distribution would likely be computationally intractable for this purpose).

3. Overall, I think this manuscript is reasonably clear/well-written: there are a few minor grammatical errors but they did not hinder my ability to comprehend the material being presented. As a minor nitpick, I would have liked to see more main paper real estate devoted to the figures, both in terms of actual size and interpretation: Figure 1(e) and (f) are a bit difficult to parse and the text doesn't provide much insight but ultimately, they did add to my appreciation of the proposed method. Similarly, the bottom-right quadrant of Figure 2 provides little compelling evidence in favor of their method.

4. My impression of this work is that it is of low to middling impact, given the somewhat niche use case of low-dimensional, non-negative batch quadrature; this is somewhat validated by the limited existing work/baselines to compare against in this space (of course, the lack of existing work is not being held against the authors, I bring it up merely as a measure of potential significance). Furthermore, related to my notes about quality, the myriad design choices represent a nontrivial barrier for other practitioners/researchers looking to experiment with/improve upon the work presented here.

---

> ### Author Response · Authors · 2022-08-02
> **Authors' response summary: added experiments of an ablation study and a sensitivity analysis, improved technical clarifications**
>
> We thank the reviewer for the helpful comments and suggestions. We have updated the manuscript to address all points raised **with red coloured text**. We address the reviewer's comments and questions as follows:
>
> C1: On model complexity and analysis.
>
> A1: **The main idea is intuitive and based on theoretical results**, as we commented to reviewer 8yqu. Also, we added an ablation study and the hyperparameter sensitivity analysis. The ablation study reveals that all component contributes to fast convergence. Hyperparameter importance analysis with functional ANOVA showed excellent correspondence to the theoretical aspects, giving clear guidelines for setting the hyperparameters $N$, $M$, and $r$.
>
> C2: Motivation on IVR, relationship with DPPs
>
> A2: **Half correct**. Mixing with the prior is necessary for convergence acceleration, not to ensure global optimisation. Thank you for finding the inaccurate description! However, the integral variance reduction (IVR) proposal distribution is mathematically motivated by Lemma 1 in the supplementary material. As Lemma 1 shows, the proposal distribution $g(x) = \sqrt{C^L_\textbf{y}(x,x)} \pi(x)$ gives the optimal upper bound to reduce the integral variance of the empirical measure with the mathematical proof. However, the square-root kernel is intractable to be sampled; thus, we linearised as Equation (8)  (When r=0.5, the IVR becomes the linearised expression of the square root). We added the comparison result with the square root and the linearised IVR in table 1 in the supplementary, showing it gives the beneficial speed up with identical accuracy.
>
> **Yes**, DPP considers the correlation correctly, whereas RCHQ assumes i.i.d. However, DPP requires prohibitive computation. Table 1 in the reference [37] compares DPP-based KQ [S4] and RCHQ ("N. + empirical" refers to RCHQ), which clearly shows that RCHQ provides not only tractable computation but also competitive theoretical bound of worst-case error with mathematical proof.
>
> C3: impression on impact, complexity
>
> A3: **Not quite**. The limited baselines are due to the inherent difficulty in parallelisation of active learning GP algorithms and are not evidence of weak impact. Rather, we believe it is the opposite. Only a handful of algorithms can handle sample-efficient parallelisation for wide applications, such as local penalisation (batch WSABI) and DPP, both of which are computationally intractable. Our approach is not only tractable but also general-purpose; applicable to any online GP algorithms, such as Bayesian optimisation [45], dynamic optimisation like control [22], and probabilistic numerics [39]. Moreover, the low-dimensional limitation could be overcome in future work because RCHQ is agnostic to the input space, allowing quadrature in manifold space. An appropriate latent variable warped GP modelling, such as GPLVM [49], could pave the way to high dimensional quadrature. However, this is out of scope in this work. In addition, BASQ is not limited to non-negative. We can handle negative integral just because $\alpha$ in WSABI can take a negative value. Non-negativity is naturally incorporated via non-negative $\textbf{y}$ in the likelihood case. Parabolic transformation is advantageous for incorporating information of non-negativity for faster convergence by making the exploration space smaller than vanilla BQ. We emphasise that BASQ is the first general framework for fast parallelisation of online GP, which is capable of combining the other methodologies harmoniously. Finally, our approach is intuitive (see A1). The added experiments give clear guidelines for practitioners and researchers.
>
> Q1: Why Nyström? Other low-rank approximation possibilities?
>
> A4: **Because Nyström is advantageous to derive convergence based on spectral decay asymptotically and theoretically.** The only requirement for the RCHQ is a finite sequence of good test functions, so finite dimensional approximations such as random Fourier features can also be used.
>
> Q2: Why WSABI-L? Other BQ modelling possibilities?
>
> A5: **Because WSABI-L is advantageous in faster sampling.** In the supplementary, we added the ablation study on BQ modelling in table 2. We compared WSABI-M, vanilla BQ, and log-GP BQ (BBQ) [54]. WSABI-L had the fastest convergence, whereas WSABI-M had slightly higher accuracy with 20 times more overhead time due to the intractable uncertainty sampling.
>
> Q3: BASQ and batch WSABI seems identical?
>
> A6: **No, they are different.** This is just because the NS methods are too poor to plot in the same range with BQs because the expensive likelihoods makes the sample size per time identical over the methods. (see section 5.2). We separated the plot with broken axis, revealing the clear difference. It turns out batch WSABI was the performant in the battery simulator. Thank you. Very low-dimensional and sharp unimodal nature of this likelihood was advantageous for biased greedy batch WSABI, as IGB superiority supports this viewpoint.

---

> > ### Comment · Reviewer_Aavt · 2022-08-08
> > **Increasing score**
> >
> > I thank the authors for their detailed feedback and extensive ablation study that they included in the most recent version of their appendix. I find myself sufficiently convinced by their response that I am increasing my score to 6; that being said, many of the inclusions also appear to have minor grammatical mistakes/typos e.g., an extraneous "the" in line 301 and a missing "the" in line 261, which I would recommend the authors fix in the camera-ready version.

---

> > > ### Author Response · Authors · 2022-08-09
> > > **Acknowledgment and minor correction of the manuscript**
> > >
> > > We are thrilled you respected our explanation! We sincerely appreciate increasing your score. We have corrected the erroneous parts you pointed out for us tentatively. We promise that we will definitely correct the grammar and expressions in the camera-ready version.

---

### Official Review · Reviewer_8yqu · 2022-07-13

**Rating:** 7
**Confidence:** 4
**Soundness:** 4 excellent
**Presentation:** 3 good
**Contribution:** 4 excellent

**Summary:**

This paper makes a contribution to speeding up Bayesian Quadrature (BQ), a popular method for computing the posterior and evidence. The proposed method builds on the weighted kernel quadrature method of Hayakawa et al. and uses the (proven) advantage of kernel recombination to improve sample selection for worst case error reduction and hence improve the accuracy of the quantities estimated by standard BQ methodology. It is shown that the method performs favourably compared to the state of the art WSABI.


**Questions:**

To me, the weakest point of the paper is a somewhat unclear treatment of the relationship between KQ and BQ, w.r.t reference [40] and the explicit contribution made. It is said that "the point configuration in KQ with a small worst-case error gives a good way to select points to reduce the integral variance in Bayesian quadrature." Now, to the best of my knowledge, (naive) kernel herding is limited by the computation of certain integrals required when sequentially selecting points. As far as I understand, the reference [37] is a practical way of circumventing this limitation? It would be nice to expand a bit more on this crucial point upon which the paper is ultimately based. It would also be good to mention some potential areas, if any, where the proposed method performs worse than existing ones.


**Limitations:**

Yes.

**Strengths And Weaknesses:**

In my view, this paper is an obvious improvement on the WSABI methodology with few drawbacks (other than those intrinsic to BQ utilizing Gaussian processes in general, such an upper bound of applicability to 10-15 dimensions) and a very natural alternative to it that I can see being used in practice. It is well-written, complete and easy to read.

---

> ### Author Response · Authors · 2022-08-02
> **Authors' response summary: improved technical clarifications**
>
> We thank the reviewer for the helpful comments and suggestions. We have updated the manuscript to address all points raised **with orange-coloured text**.
> In particular, the clarifications of the relationship between BQ and KQ, and the RCHQ superiority over herding. We address the reviewer's questions and comments as follows:
>
> Q1: As far as I understand, the reference [37] is a practical way of circumventing this limitation?
>
> A1: **Almost**. We will explain the reasons why we can parallelise the batch sampling via KQ and the faster convergence of RCHQ than kernel herding. Firstly, the trick that achieves parallelisation is the alternately subsampling in section 4.1, not RCHQ itself. While BQ aims to calculate the target integral $Z = \int \ell(x)\pi(x)dx$, RCHQ over a single iteration aims to calculate the integral of empirical measure defined by $N$ subsamples. At each iteration, we greedily select the batch candidates via RCHQ that can minimise the integral variance over the current empirical measure. As we gather more observation data points and update the kernel (GP), the above two integrals approach the same. In other words, any KQ method, including kernel herding, can be applied to the batch selection via this alternately subsampling scheme. Secondly, such a dual quadrature scheme tends to be computationally demanding, but tractable computation and superb sample efficiency of RCHQ permit scalable batch parallelisation. The reason why RCHQ converges faster than herding is that RCHQ exploits more information than herding. While herding greedily optimises sequentially, RCHQ explicitly exploits the information of the spectral decay of the kernel and the probability distribution, both of which herding neglects. Exploiting the spectral decay corresponds to capturing the approximately finite dimensionality of the kernel. RCHQ adopts the Nyström method for its approximation. This convergence rate superiority can be confirmed in figure 2(a) in the reference [37]. ("N. + emp + opt" refers to RCHQ.) While RCHQ exponentially decays, herding does not show such fast decay in the Gaussian setting. Therefore, BQ with RCHQ can converge faster than BQ with kernel herding, allowing scalable and sample-efficient batch selection.
>
> Q2: It would also be good to mention some potential areas, if any, where the proposed method performs worse than existing ones.
>
> A2: **Probably there is.** The advantage of herding over RCHQ is the computation cost. In the small batch size setting, the difference in the level of convergence between herding and RCHQ is much smaller than in the large batch size $n$. Therefore, herding might perform better than RCHQ in the small batch with a very cheap likelihood case as herding might earn more samples than RCHQ. The comparison against other KQ methods is summarised in table 1 in the reference [37]. RCHQ gives a small theoretical bound of the worst-case error with tractable computation cost compared to herding, DPP, CVS, and vanilla BQ.

---

### Author Response · Authors · 2022-08-06
**Author's summary of response, added experiments for clarification stated in the previous rebuttal**

Dear reviews, we would like to thank you again for your thorough and thoughtful review wholeheartedly. We are excited that all comments made our work significantly better. We are welcome on discussing if you have any unclear statements on our comments.

In addition to our previous rebuttal comments, we have run additional experiments for discussion to clarify the extendability of our work. We wrote that our BASQ framework could extend to other kernels, prior distributions, higher dimensions, and Bayesian optimisation. Application to Bayesian optimisation is evident because several existing works already use kernel quadrature with DPP for batch Bayesian optimisation. The merit of adopting BASQ for batch BO is the same; faster than DPP with tractable computation. This will surely benefit the autoML community, such as automatic hyperparameter tuning.

The other possibilities are inherently derived from kernel selection. When we can select arbitrary kernels, we can adopt manifold GPs, such as neural tangent kernel, for high-dimensional Bayesian Quadrature (BQ). In the existing BQ work, the kernel selection was bespoke: we needed to separately model the GP with specific kernels to derive the analytical form of integral at hand. However, RCHQ itself is not limited to a particular kernel. Therefore, we experimented with the calculation of model evidence using RCHQ, the weighted summation of the GP means, instead of an analytical form of integral exploiting Gaussianity.

We have run the experiments with seven popular kernels in Table 3 in the supplementary, showing BASQ can successfully perform quadrature for arbitrary kernels without bespoke BQ modelling. Adopting a more appropriate kernel to the likelihood of interest can improve the evidence inference, the same with conventional GP regressions. Furthermore, release from bespoke modelling permits kernel learning, such as deep kernel learning, to automate kernel selection in future work.

We appreciate the reviewers, which have significantly improved our work significance and generalisability. Reviewer 8yqu's comments improve the clarity on the connection between BQ and KQ, leading to the clarification that our BASQ framework of alternately subsampling scheme permits the adoption of other KQ methods, including herding, to batch BQ. Reviewer Aavt's comments enhanced the clarity of each component's influence via the ablation study.
Furthermore, the added commented hyperparameter sensitivity analysis provides a user-friendly guideline for practitioners and researchers who wish to improve upon our work. Reviewer CnRy's comments improved the clarifications on generalisability to select arbitrary kernels.

We will conclude with the strengthened merits of our work. Our work, BASQ, is the first to allow scalable parallelisation of active learning GP and the arbitrary kernels and KQ method selection to the parallelisation. Moreover, as BQ can solve Bayesian inference, the foundation paradigm of machine learning, we believe that the extendability of our work will benefit many related NeurIPS communities, not only in the active learning GP community but also in method development and applications of efficient Bayesian computation appeared in many ML tasks.

---

### Meta-Review · Area_Chair_Vvqu · 2022-08-30

**Recommendation:** Accept
**Confidence:** Certain

**Metareview:**

The initial round of reviews for the submitted manuscript was mostly positive in tone, but this enthusiasm was tempered by a number of deep technical issues raised by the reviewers. Fortunately, the author rebuttal and author--reviewer discussion phases went a long way toward clearing up some initial confusion and clarifying the contributions of the authors, which swayed the prevailing opinion of the reviewers toward acceptance.

I want to commend the authors for their enlightening contributions to that discussion, which assuaged most of the reviewers' initial complaints.

_However, I would also like to stress that it is critical that the fruits of this discussion be incorporated into a revised version of this manuscript._ The reviewers are unanimous in this opinion.

In particular, I direct the authors to the conversation with reviewer CnRy and the points raised about:

- the manner in which the theoretical results were initially presented in the discussion/abstract, and
- more clarity regarding the assumptions made Theorem 1 and the notation used to communicate these assumptions and the theorem.


**Award:**

No

---

### Decision · Program_Chairs · 2022-09-14

Accept